# Concurrent improvements in maize yield and drought resistance through breeding advances in the U.S.Corn Belt

Haidong Zhao [1], Jesse B. Tack [2], Gerard J. Kluitenberg[1], M. B. Kirkham [1], Gretchen F. Sassenrath [3], Lina Zhang [1], Nenghan Wan [1], Zhijuan Liu [4], Jin Zhao[4], Amanda Ashworth [5], Prasanna H. Gowda [6] & Xiaomao Lin [1] ✉

Drought increasingly challenges rainfed maize (*Zea mays* L.) production worldwide, with pressures expected to intensify under future climate scenarios. Recent studies have examined the genetic and physiological bases of yield and drought tolerance improvements in maize; however, comprehensive, field-based quantification of synchronous improvements of yield and drought resistance across diverse environmental conditions remain limited. By compiling a dataset of 92,096 hybrid-trial observations across the U.S. Corn Belt (2000–2020), our environmental index approach provides evidence of consistent yield increases across diverse environmental conditions. Using linear mixed-effects modeling, we reveal these gains are accompanied by enhanced drought resistance during the grain filling period. Projections suggest that by 2100, new hybrids could transform drought resistance, reducing yield losses by 17.8% compared to old hybrids, suggesting the potential of breeding innovations to buffer maize against drought stress. This study highlights recent breeding efforts, reinforcing adaptative capacity of maize and providing a promising pathway to sustain food security in a warming climate.

Drought continues to be a major driver of yield loss in rainfed crop production systems worldwide[1–3] and poses a substantial threat to global food security[4]. Over the past decade, drought has resulted in approximately $30 billion in global crop production losses[5]. Projected climate change is leading us toward a hotter, more parched world[6] (Supplementary Fig. 1), which will impede crop production[7,8]. For instance, in the next two decades, projected increases in drought events will decrease wheat yield by 15%[9] and, in the next century, will decrease maize yield by up to 30% in the U.S. Corn Belt[10]. These projections highlight the urgent need for effective crop adaptation strategies to ensure sustainable crop production in the face of climate change[11].

Improved agronomic management practices and enhanced breeding technologies have been traditional adaptation strategies to mitigate the influence of detrimental environmental conditions. Management practices for increased resistance to adverse climatic conditions have received much attention in the literature, with strategies including improving soil quality[12], adjusting planting dates[13], bolstering conservation agriculture[14], and increasing the use of irrigation[15]. Even though advances in breeding technology are associated with higher average yield under historical climate conditions[16–18], it remains unclear whether these advancements have increased resistance against weather risks. This is particularly relevant for maize hybrids in the U.S. Corn Belt, which annually produces an average of 0.32 billion

[1]Department of Agronomy, Kansas State University, 2004 Throckmorton Plant Sciences Center, Manhattan, KS, USA. [2]Department of Agricultural Economics, Kansas State University, Manhattan, KS, USA. [3]Kansas State University Southeast Research and Extension Center, Parsons, KS, USA. [4]College of Resources and Environmental Sciences, China Agricultural University, Beijing, Haidian District, People's Republic of China. [5]USDA, Agricultural Research Service, Poultry Production and Product Safety Research Unit, Fayetteville, AR, USA. [6]United States Department of Agriculture, Agricultural Research Service, Southeast Area, Stoneville, MS, USA. ✉e-mail: xlin@ksu.edu

metric tons of maize, equivalent to 30% of global production (Supplementary Fig. 2). Maize yield in this region is susceptible to drought stress, with roughly 13% of yield variation associated with drought[19].

In response to these challenges, breeders have enhanced drought resistance traits in maize over the past two decades primarily through improved accelerated breeding achieved by combining advanced phenotyping and genomic prediction[17,20–22]. After the widespread drought in 2012, which caused a 30% yield loss, the adoption of drought-tolerant (DT) maize hybrids by U.S. farmers substantially increased, with over 20% of the total area planted with DT hybrids in 2016[23]. Field studies have shown that DT hybrids generally exhibit lower susceptibility to drought stress compared to non-DT hybrids[17,24,25]. Furthermore, recent studies have shown that dedicated breeding and technology advancements over the past two decades have markedly enhanced the genetic gains of maize yield under water-limited conditions, increasing from 0.06 t ha⁻¹ year⁻¹ to 0.08 t ha⁻¹ year⁻¹ [22,26–28]. This progress has mainly been achieved by the selection of multiple drought-tolerance mechanisms, such as shifting the patterns of water use. While these findings are well-documented in field studies, the speed and effectiveness of breeding improvements to drought resistance across diverse environments in large-scale maize breeding trials remain poorly understood. Addressing this knowledge gap is critical for targeting future breeding strategies and providing information for research

investment policies that aim to ensure sustainable food production in the future[29,30].

While previous studies have examined temporal variation in maize sensitivity to drought using aggregated farm- or county-level yield data[19,31,32], they fail to account for information on the specific hybrids grown by farmers, which are likely to be heterogeneous, limiting the ability to assess whether breeding advancements have effectively improved drought resistance. To address these gaps, we utilized a large, field-level dataset consisting of 92,096 hybrid-trial observations across 12,847 hybrids (Supplementary Fig. 3) grown at 63 rainfed sites associated with land-grant universities in the five primary U.S. maize-producing states (Iowa, Illinois, Minnesota, Ohio, and Wisconsin) over two decades (2000–2020) (Fig. 1a). This dataset allowed us to track changes in drought resistance with hybrid advancements, providing insights into how breeding strategies have evolved to cope with drought stress.

Typically, drought studies focus on soil drying; however, even when soil retains 50–100% of its available water (no soil water stress), the plant shoot can be exposed to hot and dry air, triggering a high vapor pressure deficit (VPD)[2,33]. This atmospheric drought can significantly impact plant growth rates, leading to reductions in crop yield. VPD has been frequently used to link crop yield and drought in previous studies[2,10,34–36]. While the reproductive period of maize is the most vulnerable one to water stress[20,37], drought can also impact maize

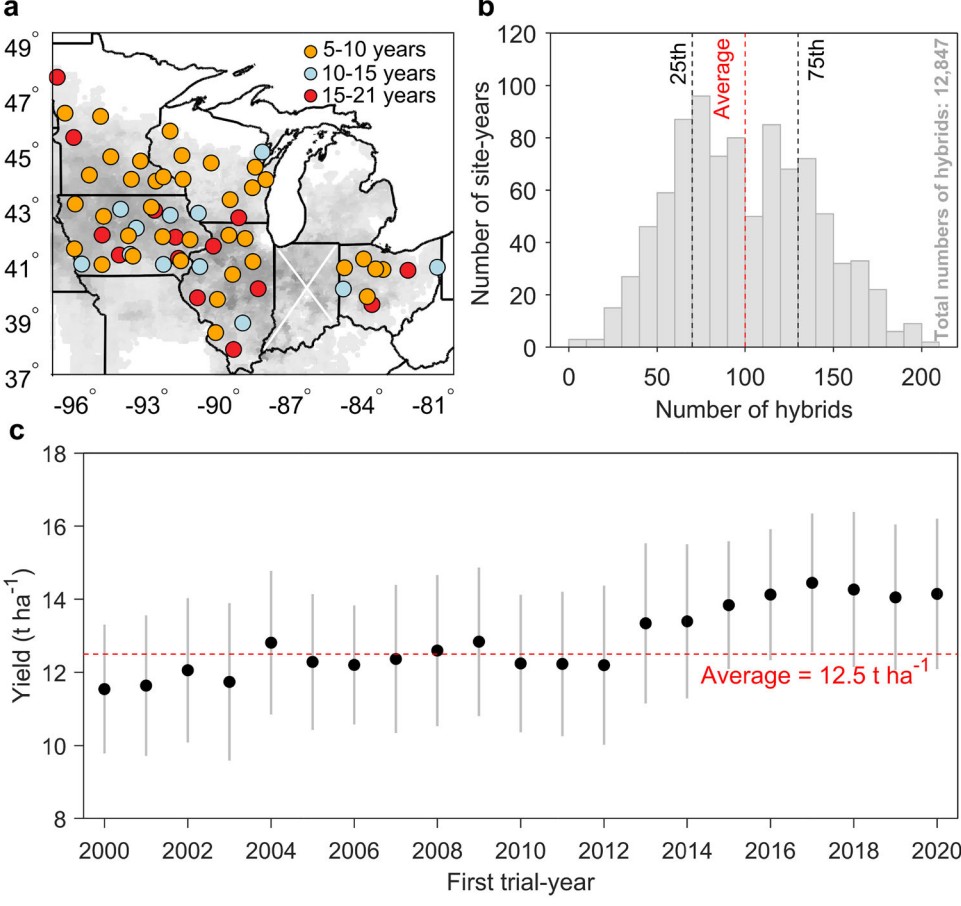

**Fig. 1 | Characterization of the field trial data. a** Map of the main study region with individual colored circles showing field trial years. The gray-shaded map background shows the main maize harvested area in the 2000s. Field-trial data are not available in Indiana (white "X"). **b** The histogram is the number of maize hybrids for each trial site-year. The 25th and 75th percentiles of maize hybrids are marked by the black dashed lines, and the average is marked by dashed red lines. **c** Time series of maize yield across years in which the hybrids first appeared in the field trial data. Each dot is the first-trial-year average yield (The calculation procedure is shown in Supplementary Fig. 26). The line indicates one standard deviation. The average sample size of each first trial year is 4386. The red dashed line shows the average yield across all hybrids. Source data are provided as a Source Data file.

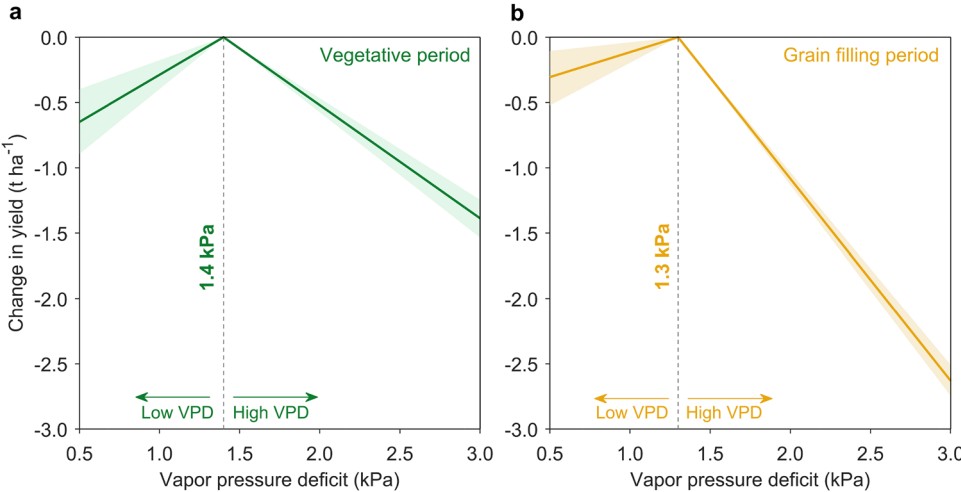

**Fig. 2 | Nonlinear effect of vapor pressure deficit (VPD) on maize yield during the growing season. a** Effect of VPD on yield during the vegetative period (from planting to silking). Slopes to the left and right of the VPD threshold (vertical dashed line) indicate sensitivity to VPD below and above the threshold. The solid lines correspond to the ensemble average with colored shadow areas displaying the 2.5th–97.5th percentile range of 1000 bootstraps. **b** Effect of VPD on yield during the grain filling period (from silking to maturity), with sensitivity estimated below and above the VPD threshold as in (**a**). Source data are provided as a Source Data file.

yield at other stages of the growing season[2,38]. Therefore, we focus on the effect of VPD throughout the growing season, considering specific phenological periods derived from field observations and data from the United States Department of Agriculture's National Agricultural Statistics Service (USDA-NASS) (Supplementary Text 1 and Supplementary Figs. 4 and 5). We leverage these data to examine how and to what extent recent changes in high-yielding (HYH), median-yielding (MYH), and low-yielding hybrids (LYH) (Supplementary Text 2 and Supplementary Fig. 6) are associated with the effect of drought on maize yield. Using a linear mixed-effects model, we estimate how drought resistance in maize has evolved over two decades of breeding efforts. Finally, we project the potential impact of future climate change on maize yield, comparing hybrids from different breeding periods to assess the impacts of recent advancements in drought tolerance under projected climate scenarios.

## Results
### Historical effect of drought on maize yield
Maize yield generally exhibits a threshold response to vapor pressure deficit (VPD) in the field. Beyond this threshold, partial stomatal closure can be triggered, potentially limiting yield, whereas below it, hybrids would be insensitive to VPD[36,39]. However, the VPD threshold for maize during specific phenological periods remains unclear in the literature, as these thresholds can vary depending on the methods used to calculate VPD and the specific growth stages considered. To address this, we adopted a piecewise linear approach based on daily weighted VPD (see "Methods", Supplementary Text 3, and Supplementary Fig. 7), allowing for distinct thresholds for the vegetative period (VEG; from sowing to silking) and grain filling period (GFP; from silking to maturity). We estimated the performance of the regression model over a range of possible threshold combinations and selected the optimal thresholds based on the best model fit (minimizing the Akaike Information Criterion; Supplementary Fig. 7). Our analysis revealed optimal VPD thresholds of 1.4 kPa during the VEG period and 1.3 kPa during the GFP period (Fig. 2).

Our results demonstrated a nonlinear relationship between maize yield and VPD at these two phenological periods. Specifically, the average sensitivities (Fig. 2a and Supplementary Table 1) revealed that an increase in VPD had a significant positive effect ($p < 0.001$, based on bootstrap statistics) on maize yield (0.72 t ha$^{-1}$ kPa$^{-1}$) during the VEG

period at low VPD levels (below 1.4 kPa), and a strong negative yield effect (−0.87 t ha$^{-1}$ kPa$^{-1}$, equivalent to 6.96% of historical yield (Fig. 1c)) ($p = 0.003$) at higher VPD levels (above 1.4 kPa). Similarly, during the GFP period, a 1 kPa increase in VPD below the threshold (1.3 kPa) was significantly associated ($p < 0.001$) with a yield improvement of 0.38 t ha$^{-1}$, but was markedly associated ($p < 0.001$) with a yield decrease of 1.55 t ha$^{-1}$ (equivalent to 12.4% of historical yield) at high VPD levels (above 1.3 kPa) (Fig. 2b and Supplementary Table 1). This highlights that the effect of drought on yield during the grain filling period, characterized by VPD above the threshold, is greater compared to the effect of drought during the VEG period, consistent with a previous study[38]. We also found a similar association between VPD and maize yield for three yield types (LYH, MYH, and HYH; Supplementary Fig. 8 and "Methods"). To test the robustness of our findings, we used three additional near-optimal threshold combinations (blue cross; Supplementary Fig. 7). The nonlinear effect of maize yield to VPD remained consistent (Supplementary Fig. 9). This nonlinear response was also robust (Supplementary Fig. 10), even after including data from the western U.S. corn belt (Supplementary Fig. 11), a region typically characterized by drought-prone environments. The result highlights the detrimental impact of high VPD on maize yields, consistent with a previous study[40]. Our analysis also revealed that all estimated response functions relating to maize yield and precipitation displayed a concave shape during the VEG and the GFP periods for all three yield types (Supplementary Fig. 12). This suggests that increased precipitation during these growth stages improves maize yield under dry conditions and decreases yield under wet conditions.

**Resistance to drought resulting from recent breeding progress**
To track yield progress under varying environmental stress levels, a precise measure of stress experienced by crops is key. This measure is often quantified as average yield across all hybrids grown at a given site-year, known as the environment index (EI)[41,42]. Higher EI represents environments with diminished environmental stress. Here, we calculated the EI for each site-year across the three yield types (Supplementary Text 4 and "Methods"). We found that the mean yield for each percentile of the EI exhibited significantly positive trends ($p < 0.05$; average rate of 0.15 t ha$^{-1}$ yr$^{-1}$; 95% confidence interval: 0.13–0.17 t ha$^{-1}$ yr$^{-1}$; degrees of freedom = 19) during 2000–2020 across the three yield types (Fig. 3a–c). Although yield

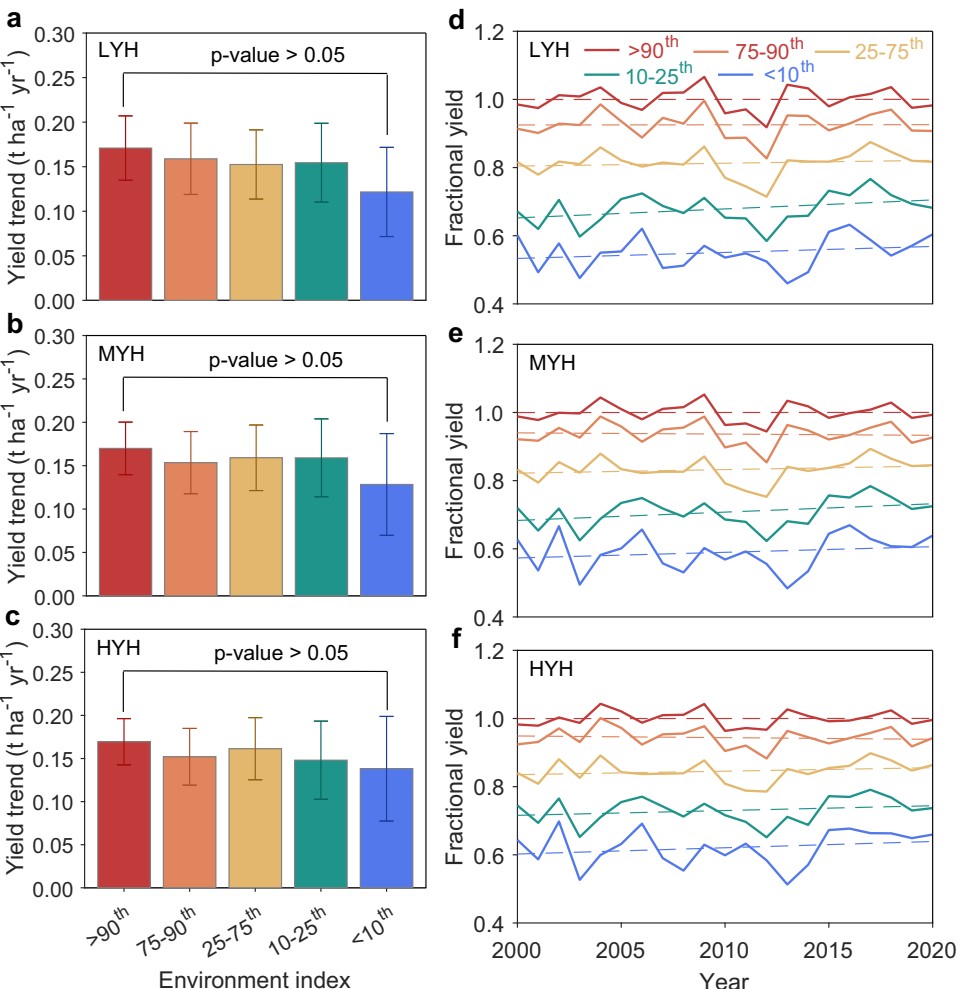

**Fig. 3 | Trend of maize yield across five environmental index levels from 2000 to 2020 for low-yielding hybrids (LYH), median-yielding hybrids (MYH), and high-yielding hybrids (HYH).** a–c Trend of maize yield across five environmental index levels from the highest environmental index level (red color) representing the best yield conditions, to the lowest environmental index level (blue color) indicating the worst yield conditions. Bars refer to the mean yield trend, and error bars are at the 95% confidence interval (sample size for each environmental condition = 21 years). Trends were estimated using linear regression (two-sided test).

No adjustment for multiple comparisons was applied. The statistical significance of the difference in yield trends between the best- and lowest-yielding conditions was tested using analysis of covariance (ANCOVA). d–f Time series of maize yield (relative to the best environmental condition) for LYH, MYH, and HYH, respectively (see "Methods"). The solid lines show the mean fraction yield, and dashed lines indicate fitted linear trends for five environmental index levels. Source data are provided as a Source Data file.

trends under the lowest stress level (EI >90th percentile; 0.17 t ha$^{-1}$ yr$^{-1}$, equivalent to rate of 1.36% yr$^{-1}$) marginally surpassed that under the highest stress level (EI <10th percentile; 0.13 t ha$^{-1}$ yr$^{-1}$, equivalent to rate of 1.12% yr$^{-1}$), the difference was not statistically significant ($p > 0.05$; average difference −0.04 t ha$^{-1}$ yr$^{-1}$; 95% confidence interval: −0.10 to 0.02 t ha$^{-1}$ yr$^{-1}$; degrees of freedom = 38) (Fig. 3a–c), indicating parallel yield gains over breeding across a range of environments. The mean annual yield trend across diverse environments is a 1.2% increase per year, which aligns with previous findings in the literature[17]. By expressing maize yield as a percentage relative to yearly expected yields under the lowest stress level, we observed no decrease in yield trends even under the most stressed level (Fig. 3d–f). This suggests that recent breeding progress has consistently increased maize yield under both favorable and stressed environmental conditions. This finding was also robust using alternative thresholds of the environment index (Supplementary Fig. 13).

To specifically examine how breeding progress influences the resistance of maize yield to drought, characterized by VPD above the threshold, we employed a varying-slope multilevel model (see "Methods") that accounts for interactions between hybrids and high VPD

during the vegetative and grain filling periods. This statistical model has been employed in estimating the effect of breeding progress on climate change adaptation of other crops, such as winter wheat[43] and sorghum[44]. We found no statistically significant change in maize yield resistance to drought during the VEG period ($p > 0.10$; mean effect size: −0.01 t ha$^{-1}$ kPa$^{-1}$ yr$^{-1}$; 95% confidence interval: −0.06 to 0.04 t ha$^{-1}$ kPa$^{-1}$ yr$^{-1}$; degrees of freedom = 19), but a significant improvement ($p < 0.001$) at a rate of 0.12 t ha$^{-1}$ kPa$^{-1}$ yr$^{-1}$ (95% confidence interval: 0.07–0.17 t ha$^{-1}$ kPa$^{-1}$ yr$^{-1}$; degrees of freedom = 19) during the GFP period across maize hybrids released between 2000 and 2020 under the MYH yield type (Fig. 4a, b), which aligns with the results estimated for LYH and HYH yield types (Supplementary Fig. 14). This remained robust when using three additional near-optimal thresholds combinations (Supplementary Figs. 7d and 9c, f, i). Furthermore, by expanding our analysis to include data from the western U.S. corn belt, we found a consistent improvement in drought resilience of maize yield over breeding progress (Supplementary Fig. 15). We also found a distinct break in the increasing resistance of maize yield to drought during the GFP period for hybrids released after 2012 (2012–2020; Fig. 4a), a period during which more drought tolerant

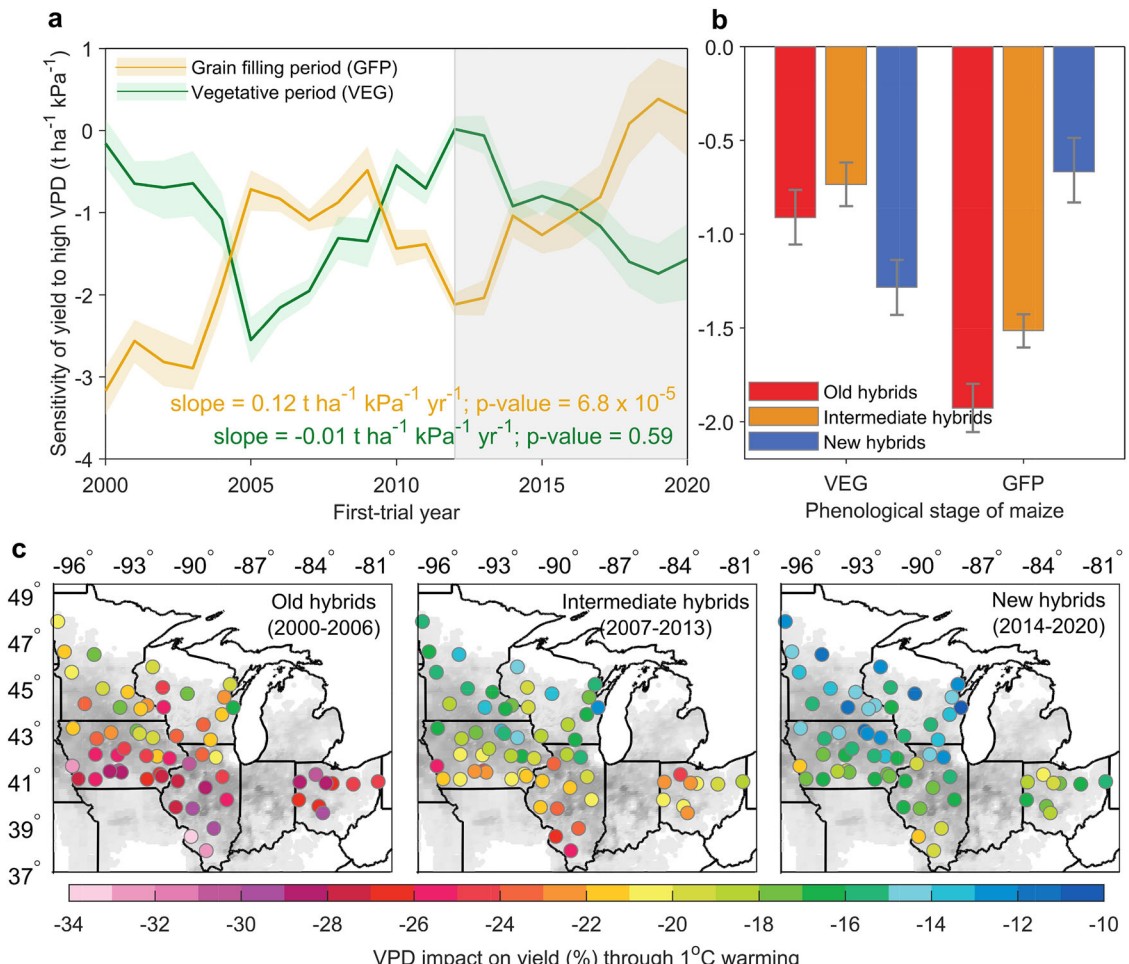

**Fig. 4 | Change in sensitivity of maize yield to vapor pressure deficit (VPD) during the growing season for median-yielding hybrids. a** Time series in sensitivity of yield to VPD over breeding during the vegetative period (VEG; from sowing to silking) and grain filling period (GFP, from silking to maturity). The solid lines represent the ensemble average with colored shadow areas displaying the 2.5th–97.5th percentile range of 1000 bootstraps. The filled area represents the appearance of drought-tolerant (DT) hybrids. Trends were estimated using linear regression (two-sided test). No adjustment for multiple comparisons was applied. **b** The sensitivity of yield to VPD for old hybrids (2000–2006), intermediate hybrids (2007–2013), and new hybrids (2014–2020). Bars represent the ensemble average with error bars showing the 2.5th–97.5th percentile range of 1000 bootstraps. **c** The spatial effect of VPD change on yield for the three hybrid age types through 1 °C warming during the growing season, including the VEG and GFP periods. Source data are provided as a Source Data file.

(DT) hybrids were gradually introduced (Supplementary Fig. 16a). This implies the introduction of DT hybrids may directly enhance drought resistance. We explored this possibility by comparing the drought resistance of DT and non-DT hybrids (see "Methods"). The technologies of DT hybrids in our data included Pioneer Optimum AQUAmax and DroughtGard products. Although drought resistance was similar between the two types of hybrids during the VEG period, the drought resistance of DT hybrids was 30% higher (significantly at 95% confidence interval) than that of the non-DT hybrids during the GFP period (Supplementary Fig. 16b). To confirm our findings, we also re-developed the drought index using daily mean VPD and re-estimated the response of hybrid advancements in drought resistance. The result (Supplementary Fig. 17) was consistent with that using daily weighted VPD (Fig. 4a). Additionally, we assessed how breeding advancements may have influenced drought resistance of maize yield under different production conditions. We categorized trial years into three production conditions for each state: good, normal, and bad years (see Supplementary Text 5 and Supplementary Fig. 18). We found no statistically significant change ($p > 0.10$) in drought resistance during the VEG stage (Supplementary Fig. 18), consistent with our findings (Fig. 4a). However, during the GFP period, a clear gradient emerged:

VPD sensitivity did not significantly change in good years ($p > 0.10$) but showed a statistically significant improvement in bad years ($p = 0.01$), and a marginally significant increase in normal years ($p = 0.06$) (Supplementary Fig. 18). These findings suggest that breeding progress over the past two decades has improved drought resistance during the GFP period, especially under unfavorable production conditions. This result aligns with prior work[28], which showed that sequential improvements in hybrid yield under drought were driven by breeding-induced shifts in water-use patterns. Such shifts reduced hybrid sensitivity to drought during the phase of flowering to grain filling.

We also computed the net effect of increasing VPD, resulting from 1 °C warming, on maize yield for old (2000–2006), intermediate (2007–2013), and new (2014–2020) hybrids at each field site (see "Methods"). For all three hybrid age groups, an increase in VPD was consistently associated with a decline in maize yield across all field sites, but yield loss was greater in the southern region, owing to more frequent exposure to high VPD (Fig. 4c). Importantly, the effect of increasing VPD on yield was alleviated with breeding progress, including DT introductions. Specifically, the predicted yield loss due to increased VPD was 25% on average for old hybrids, but it was reduced to 16% for new hybrids (Fig. 4c).

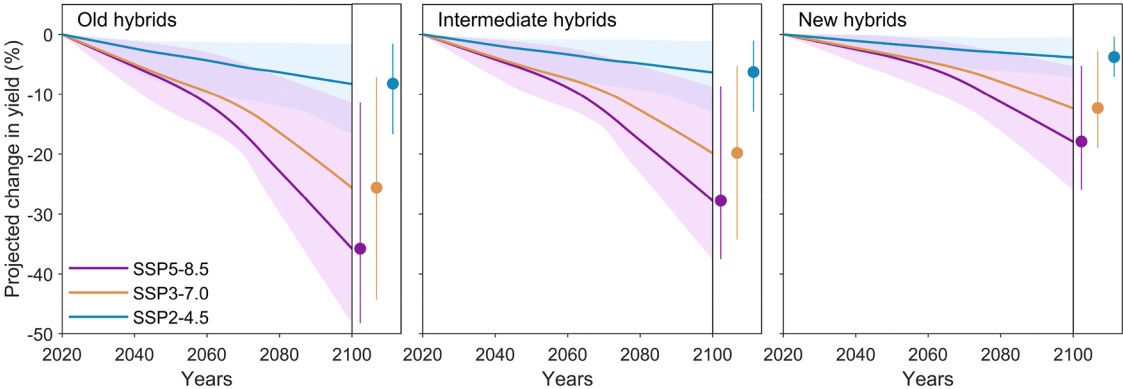

**Fig. 5 | Projected yield change driven by future vapor pressure deficits over time relative to the historical average for old (2000-2006), intermediate (2007–2013), and new (2016–2020) hybrids of the median-yielding type under three climate scenarios.** The solid lines highlight average projections with the colored shadow areas displaying the 2.5th–97.5th percentile range of 1000 bootstraps × 7 climate models (sample size = 7000). The dots are average, and the error bars show the 2.5th–97.5th percentile range of projected yield impacts by 2100. Source data are provided as a Source Data file.

## Potential impacts of drought on maize yield amid projected climate change

To estimate potential impacts of projected future VPD (Supplementary Fig. 1) on maize yield, our analysis considered old, intermediate, and new age hybrids (Fig. 4b). We utilized projected climate conditions up to 2100 based on three shared socioeconomic pathways (SSPs) (SSP2-4.5, SSP3-7.0, and SSP5-8.5) from seven climate models in the Coupled Model Intercomparison Project Phase 6 (see "Methods"). To refine the projection, we formulated our estimations by considering projected VPD, aligning them with the established impact of VPD on past yield outputs. We accounted for uncertainty both in the relationship between maize yield and VPD and future climate projections (see "Methods"). Based on historical mean phenology (Supplementary Fig. 19), we projected that yield loss for old hybrids would continue to increase, reaching up to 8.3% (95% confidence interval: 1.6%–16.7%) in SSP2-4.5 and 35.8% (95% confidence interval: 11.4%–48.2%) in SSP5-8.5 by the end of the century (Fig. 5), relative to the historical average yield (12.5 t ha$^{-1}$; Fig. 1c). For new hybrids, the projected yield loss showed only a moderate increase in the SSP2-4.5 with an average yield loss of 3.8% (95% confidence interval: 0.5%–7.2%), while projected yield loss would reach up to 18.0% on average (95% confidence interval: 5.3%–26.0%) in SSP5-8.5 (Fig. 5). The projections for yield loss suggest that new hybrids exhibit greater potential resistance to the future effect of VPD relative to old hybrids, but they also indicate that drought would still limit yield even for new hybrids (Fig. 5). Given the potential for future warming to drive earlier sowing dates and faster plant development rates, we also estimated the effect of VPD on yield when artificially advancing the phenological periods by 10 days. We found that the projected effect of VPD on yield (Supplementary Fig. 20) was slightly lower compared to results obtained using the historical average phenology (Fig. 5).

## Discussion

This study measured the resistance of maize yield to drought resulting from breeding improvements over the past 21 years using rainfed, university field-trial data from the U.S. Corn Belt. Our results provide evidence that maize yield declines when VPD surpasses a threshold during the growing season (Fig. 2), and that breeding progress over the past two decades has consistently improved yield regardless of either favorable or stressed environmental conditions (Fig. 3). We also find evidence that recent breeding efforts have produced hybrids that are more resistant to drought through increasing drought tolerance during the grain filling period (Fig. 4). Additionally, we projected that, under the worst climate scenario (SSP5-8.5) and relative to the historical climate, maize yield would decrease 11.4%–48.2% for older

hybrids, while this effect can be alleviated to 5.3%–26.0% when using newer hybrids (Fig. 5).

The increased resistance of maize yield to drought (Fig. 4a, b) can be attributed to several key factors. First, the introduction of drought tolerant (DT) hybrids into our dataset since 2012 has enhanced drought resistance (Supplementary Fig. 16). This is consistent with previous studies[24,25], which have shown that targeted breeding for drought tolerance improves yield stability and resilience under water-limited conditions. These DT hybrids likely played a crucial role in improving drought adaptation, particularly during the reproductive stage of crop development.

A second plausible factor contributing to improved drought resistance is the incorporation of the limited-transpiration trait[36,39,45], which optimizes water use by reallocating water from vegetative to reproductive growth stages. This trait improves plant water status during flowering and supports greater water use during the post-flowering period[17]. Our findings support this mechanism, as the improvements in drought resistance were most pronounced during the reproductive period (Fig. 4a, b). This underscores the importance of breeding for traits that enhance plant water use efficiency at critical growth stages.

The rise in use of genetically engineered (GM) maize hybrids, developed with traits like herbicide tolerance and insect resistance, may be another factor in increasing the resistance of maize to drought, as a result of improved plant health[46,47]. Although these traits do not directly impact the plant's ability to extract water, they reduce competition from weeds and protect plants from insect damage, thereby enhancing root development[48]. The stronger root systems improve water extraction from the soil during drought[49], while the pest resistance from GM traits can further increase resilience by preventing pest-related damage, thus reducing drought-related yield losses[32].

Beyond the plausible factors mentioned above, phenotypic changes in maize hybrids, such as increased leaf rolling in response to high VPD during the flowering period, have been observed to facilitate water conservation and increase ear size and kernel numbers in later growth stages[50]. Additionally, newer hybrids tend to manifest a lower rate of kernel abortion and barrenness[51] as well as delayed senescence[17,36] under water deficit conditions. These phenotypic changes likely contribute to greater yield stability under drought stress, particularly during the later stages of crop development.

Overall, our results indicate two important conclusions. First, maize yield has increased in parallel across diverse levels of environmental stress over the past two decades of breeding progress. Second, the resistance of maize yield to drought has been enhanced with hybrid advancements, particularly due to enhanced drought tolerance

during the reproductive stage. These findings are also consistent with previous studies[18,52], which emphasized the importance of long-term selection through integrated hybrid-by-management technology combinations in sustaining maize productivity gains and reducing hybrid sensitivities to environmental conditions. However, our findings also underscore the need for accelerated breeding efforts to offset potential future yield losses from intensifying drought conditions caused by climate change. Our results can serve as a guide for breeders developing maize hybrids, enabling more strategic improvements in their programs. Moreover, understanding the evolving impact of genetic modifications on drought resistance can be crucial for crop-yield modelers in predicting yield shocks and assessing the risk of crop failure in both current and future climates[53].

A limitation of our projections of future climate change impacts is that they are based on static historical relationships between maize yield and VPD. While this approach provides valuable insights, highlighting continuous threats of atmospheric drought in the future may oversimplify the complex, evolving interactions likely to occur under future conditions, particularly given the inherently dynamic nature of the interaction between crop trait improvement and environmental change[54]. Future studies should prioritize developing models that incorporate evolving crop traits and breeding strategies to more accurately assess future climate impacts. Additionally, our analysis does not account for the potential effect of $CO_2$ fertilization on maize yields. Although maize is less responsive to elevated $CO_2$ concentrations in terms of direct yield gains due to the C4 photosynthesis pathway, elevated $CO_2$ can enhance maize performance indirectly by reducing stomatal conductance, thereby improving water-use efficiency, particularly under drought-prone conditions[55]. Moreover, we emphasize the necessity for further research that explores the differential responses of specific maize hybrids to drought and potential variations in these responses across different locations.

## Methods

### Data source

Maize hybrid data, including observed hybrid-specific yields (adjusted to 15.5% moisture content) and phenological dates (planting and harvest), were collected and digitized from rainfed, university field performance tests of maize hybrids from 2000 to 2020 for five main production states, including Iowa, Illinois, Minnesota, Ohio, and Wisconsin, accounting for 50% of the U.S. maize harvested area (Fig. 1a). In total, the database we used included 92,096 data points across 63 field sites. The sources of all hybrids were provided by U.S. companies. The original purpose of these trials was to evaluate the performance of advanced hybrids under diverse environmental conditions, aiming to identify resilient hybrids suitable for release to farmers. Trials were conducted under "optimal" rainfed management conditions, employing site-specific agronomic treatments to optimize nutrients and minimize disease and other stresses. We segmented maize phenological periods into the vegetative growth period (VEG, from sowing to silking) and the grain filling period (GFP, from silking to maturity). Based on reported planting and harvest dates, we used state-level phenological data (Supplementary Fig. S21) from the United States Department of Agriculture's National Agricultural Statistical Service's Crop Progress Report to estimate silking and maturity dates for each trial site. Details are shown in Supplementary Text 1. To assess the impact of drought on different yield levels in subsequent analyses, we divided the maize hybrids of each site-year into three yield categories based on percentile thresholds (A case using hybrid yield data in 2000 at Belleville, Illinois for dividing maize hybrids into three yielding types is shown in Supplementary Fig. 22). These categories encompass high-yielding hybrids (HYH; >75th percentile of all hybrid yields for a given site-year), median-yielding hybrids (MYH; 25th percentile ≤ MYH ≤ 75th

percentile), and low-yielding hybrids (LYH; <25th percentile) (Supplementary Text 2 and Supplementary Fig. 6).

Daily surface observational weather data, including maximum ($T_x$; °C), minimum ($T_n$; °C), and dew point ($T_d$; °C) temperatures, along with precipitation (Prcp; mm), were derived from the Integrated Surface Dataset (ISD)[56] and interpolated with a Delaunay Triangulation[57] and applied to the field trial sites (Fig. 1a) to approximate the daily weather experienced by the crop. We used the ISD data due to its high-quality measurement of $T_d$, which is a critical weather variable in the calculation of VPD. Daily weighted VPD (kPa) to reflect the daily pattern of transpiration rate was calculated as follows[58–60]:

$$e_s = \alpha \times \left[ 0.611 \times \exp(\frac{17.3 \times T_x}{T_x + 237.3}) \right] + (1 - \alpha) \times \left[ 0.611 \times \exp\left( \frac{17.3 \times T_n}{T_n + 237.3} \right) \right] \tag{1}$$

$$e_a = 0.611 \exp\left( \frac{17.3\, T_d}{T_d + 237.3} \right) \tag{2}$$

$$VPD = e_s - e_a \tag{3}$$

where $e_s$ is daily weighted saturation vapor pressure (kPa), assuming the daily $e_s$ should be integrated from about 0900 h to evening, when net radiation becomes negative[58]. $\alpha$ equals 0.75, accounting for the daytime fraction. $e_a$ refers to actual vapor pressure (kPa) for a given daily dew point temperature ($T_d$).

### Yield gains with breeding progress

To assess yield gains due to breeding progress under various environmental stress levels for three yield types (HYH, MYH, and LYH), we calculated the environment index (EI) for each site-year, providing an overall environmental level in field trials. This approach, widely used in plant breeding, enables the assessment of crop variety/hybrids' adaptation across diverse environmental conditions by offering a simplified yet integrative measure of complex natural environments. Although this approach does not pinpoint specific environmental stressors or yield-reducing mechanisms, such as poor establishment, pest and disease, lodging, or abiotic stresses, it remains a practical tool for capturing patterns of crop performance across varying levels of integrated environmental stress. In terms of each yield type, the environment index is defined as the average yield across all hybrids grown at a given site-year[41], providing a reliable measure of stress level in field trials. We then categorized the EI of all trial sites for each year into five percentile intervals: ≤10th, 10th–25th, 25th–75th, 75th–90th, and ≥90th and calculated average yields. These intervals signify different degrees of environmental stress, with the smallest interval representing the most stressed and the highest interval representing the least stressed. It is important to note that we established the threshold of each percentile separately based on all sites in a specific year, which eliminates variations in technology and management between bins within each year. Next, we estimated yield trends for each environmental stress level using standard least-squares linear regression. The calculated process is shown in Supplementary Text 4 and Supplementary Fig. 23. To assess whether the yield trend over breeding shows a statistically significant difference across varying environmental conditions, we performed a two-sided Student's $t$-test at a 95% confidence level. A $p$-value greater than 0.05 would suggest that breeding progress contributes to relatively parallel gains in maize yield across a range of environmental conditions. We also tested other threshold intervals (20th, 40th, 60th, and 80th) to estimate yield gains of maize hybrids (Supplementary Fig. 13). To further explore this point, we analyzed relative yields, calculated as the ratio of the actual yields at specific environmental levels to the trend yields under the most favorable conditions, which serves as a

reference baseline (Supplementary Fig. 24). If breeding progress had disproportionately favored higher-yielding environments, we would expect declining trends (negative slopes) in fractional yields under stressed conditions.

## Statistical yield model

Maize hybrids typically exhibit a threshold response to VPD, where partial stomata closure occurs above a certain VPD threshold to reduce water loss under high VPD conditions[36,45]. Below this threshold, VPD tends to be positively associated with yield change. To account for this physiological factor, we developed two indices of VPD: one for the average intensity of VPD below the threshold ($VPD_b$) and the other for the average intensity of VPD above the threshold ($VPD_a$), calculated during specific phenological periods ($p$) using daily weighted VPD data.

$$VPD_{b,p} = \frac{\sum_{i=1}^{n} VPD_{i,p}}{N_p}; VPD_{i,p} = \begin{cases} VPD_{i,p}, VPD_{i,p} \le VPD_p^* \\ VPD_p^*, VPD_{i,p} > VPD_p^* \end{cases} \quad (4)$$

$$VPD_{a,p} = \frac{\sum_{i=1}^{n} VPD_{i,p}}{N_{VPD_{i,p} > VPD_p^*}}; VPD_{i,p} = \begin{cases} 0, VPD_{i,p} \le VPD_p^* \\ VPD_{i,p} - VPD_p^*, VPD_{i,p} > VPD_p^* \end{cases} \quad (5)$$

where $VPD_{i,p}$ is the daily VPD value ($VPD_i$) during phenological period $p$, with $p = 1$ for the VEG period and $p = 2$ for the GFP period. $N_p$ is the total number of days ($N$) within period $p$, and $VPD^*$ is the threshold of VPD. $N_{VPD_{i,p} > VPD^*p}$ refers to the number of days during period $p$ when daily VPD exceeds the threshold of VPD. Note that the index $VPD_a$ primarily captures the average intensity of VPD above the threshold but does not account for the frequency of VPD above the threshold, both of which are crucial factors in determining crop yield change. To account for this factor, we further adjusted $VPD_{a,p}$ by incorporating the frequency of VPD above the threshold during the phenological period, as follows,

$$\widehat{VPD}_{a,p} = VPD_{a,p} + VPD_{a,p} \times \frac{N_{VPD_{i,p} > VPD_p^*}}{N_p} \quad (6)$$

where $\widehat{VPD}_{a,p}$ represents the adjusted VPD above the threshold. $\frac{N_{VPD_{i,p} > VPD_p^*}}{N_p}$ reflects the frequency of days when VPD exceeds the threshold, relative to the total number of days in the period.

We defined 1.4 kPa and 1.3 kPa as the VPD thresholds for the VEG and GFP periods, respectively. These thresholds were selected based on optimal model performance (Eq. (7)), as determined by the smallest Akaike Information Criterion (AIC; Supplementary Fig. 7) using the full dataset. The AIC is a statistical measure used to assess the goodness of fit of different models, with a lower AIC value indicating a better fit[61]. To identify the optimum VPD thresholds, we systematically tested a range of values from 1.3 kPa to 1.9 kPa with a step size of 0.1 kPa based on the distribution of daily VPD during each phenological stage (Supplementary Fig. 7a). This range represents the 50th to 90th percentiles of daily VPD, resulting in 49 combinations of VPD thresholds (7 values for VEG period × 7 values for GFP period). For each threshold combination, model fitting was conducted to assess performance, allowing us to identify the threshold that produced the best model performance (Supplementary Fig. 7). Details regarding the selection of the threshold are shown in Supplementary Text 3. The thresholds VPD identified in this study are slightly lower than those found in a previous study[39], likely due to the

difference in calculating actual vapor.

$$Y_{i,l,t,s} = \alpha_i + \alpha_l + \alpha_{t,s} + \sum_{p=1}^{2} (\beta_{1,p} Prcp_{i,l,t,s,p} + \beta_{2,p} Prcp_{i,l,t,s,p}^2$$
$$+ \beta_{3,p} VPD_{b,i,l,t,s,p} + \beta_{4,p} VPD_{a,i,l,t,s,p}) + \varepsilon_{i,l,t,s} \quad (7)$$

where $Y_{i,l,t,s}$ is maize yield for hybrid $i$ at location $l$ in trial-year $t$ for a specific state $s$. The first three terms ($\alpha_i$, $\alpha_l$, $\alpha_{t,s}$) are effects across hybrids, locations, and a specific state-year group. This approach is commonly used to control unobserved factors (e.g., fertilizer use, soil quality) that might influence yield. $\beta_1$, $\beta_2$, $\beta_3$, and $\beta_4$ refer to the sensitivity of precipitation, precipitation squared, and VPD below and above the threshold for a specific phenological period. The squared precipitation term is included to capture the nonlinear effect of precipitation on yield. The subscript $p$ refers to the phenological period, including the vegetative period and grain filling period. The $\varepsilon$ is the error term. We quantified uncertainty in the historical maize yields-weather variables relationship by a bootstrap approach (1000 times, sampling with replacement)[62]. Our model used actual yield data instead of log-transformed yield data because the model using actual yield data had more explanatory power ($R^2 = 0.63$; Supplementary Table 1) and more normally distributed residuals (Supplementary Fig. S25). After identifying the optimum threshold of VPD, we used Eq. 7 to estimate the VPD effect on yield for three hybrid yield types (HYH, MYH, and LYH) (Fig. 2 and Supplementary Fig. 8). We also assessed whether drought-tolerant (DT) hybrids, characterized by Pioneer Optimum AQUAmax and DroughtGard products[25], improve drought resistance compared to non-DT hybrids. Specifically, maize hybrids were classified into DT and non-DT types, and the interaction terms between $VPD_a$ and hybrid types (DT and non-DT) were incorporated into Eq. 7 to estimate the sensitivity of $VPD_a$ for these two hybrid types (Supplementary Fig. 16 and Supplementary Table 2).

To investigate heterogeneous effects of VPD above the threshold ($VPD_a$) on hybrids due to breeding progress, we modified the statistical model (Eq. 7) to allow the effect of $VPD_a$ to vary across the years in which the hybrids first appeared in the trials (first trial-year; $fty$) (Fig. 4a). This was achieved by including the interaction term between $VPD_a$ and dummy variables for first trial-year. The specification of the model is,

$$Y_{i,fty,l,t,s} = \alpha_i + \alpha_l + \alpha_{t,s} + \sum_{p=1}^{2} (\beta_{1,p} Prcp_{i,l,t,s,p} + \beta_{2,p} Prcp_{i,l,t,s,p}^2$$
$$+ \beta_{3,p} VPD_{b,i,l,t,s,p} + \beta_{4,fty,p} VPD_{a,i,fty,l,t,s,p}) + \varepsilon_{i,fty,l,t,s} \quad (8)$$

## Effect of VPD change on maize yield through 1 °C warming

To better reflect temporal and spatial changes in yield driven by climate warming and hybrid advancements, we categorized maize hybrids into three age groups based on the first-trial-year, representing old (2000-2006), intermediate (2007–2013), and new (2014–2020) hybrids. The sensitivity of maize yield to VPD for each group was then estimated by adding interaction terms between hybrid groups and $VPD_a$ into Eq. 7 (Fig. 4b). To estimate the effect of VPD change on maize yield for each hybrid group under 1 °C warming, we artificially raised observed temperature by 1 °C on each day during the growing season, including the VEG and GFP periods, for each location-year (63 locations × 21 years). VPD indices were then recomputed for each phenological period. Based on the calculated VPD values, we used the regression model to predict the yield. The effects of VPD change were summarized for each location as the

average across all years (2000–2020) (Fig. 4c), as follows,

$$\%Y_l = \frac{\frac{1}{n} \times \sum_{yr=2000}^{2020} (\widehat{Y_{2,yr,l}} - \widehat{Y_{1,yr,l}})}{\hat{Y}} \times 100 \qquad (9)$$

where *%Y* represents the percentage change in yield relative to the mean observed yield ($\hat{Y} = 12.5$ t ha$^{-1}$; Fig. 1c) for each location *l*. *n* refers to the number of years, which equals 21. $\hat{Y}_1$ and $\hat{Y}_2$ are the predicted yields using the original and 1 °C-warming VPD values, respectively.

### Projected impact of VPD on maize yield

To investigate how VPD during the maize growing season would change in the future and its influence on yield, we downloaded the climate projections from the NASA Earth Exchange-Global Daily Downscaled Projections-Coupled Model Intercomparison Project Phase 6 (NEX-GDDP-CMIP6)[63], using seven climate models (EC-Earth3, INM-CM4-8, MPI-ESM1-2-HR, MPI-ESM1-2-LR, MRI-ESM2-0, NorESM2-LM, and NorESM2-MM) under three climate scenarios (SSP2-4.5, SSP3-7.0, and SSP5-8.5). The SSP2-4.5 scenario is associated with a nominal radiative forcing level of 4.5 W m$^{-2}$ anticipated by the year 2100. The SSP3-7.0 scenario is a medium-high reference scenario within the "regional rivalry" socioeconomic family with a high emission and CO$_2$ doubled by the year 2100, while the SSP5-8.5 marks the upper edge of the SSP scenario spectrum with a high reference scenario in a high fossil-fuel development world throughout the 21st century. We then interpolated these projections to each field site with a Delaunay Triangulation[57] to calculate daily VPD. Because projections for dew point temperature were not available, we used specific humidity, which describes the mass of water vapor present in a unit mass of moist air, to calculate actual vapor pressure[64]. We cannot know future maize phenological periods for certain; thus, an average of observed historical phenological periods (Supplementary Fig. 19) was used to calculate VPD. Using a fixed phenology allows for a more accurate estimation of the impact of climate change on yield by isolating the effects of phenological changes. However, given the potential for future warming to drive earlier sowing dates and faster plant development rates, we also recalculated daily VPD for the future by artificially advancing the phenological periods by 10 days. Finally, VPD derived from the seven climate models, along with the sensitivity of maize yield to VPD, were used to estimate the average yield effects. All data sources used in this study are shown in Supplementary Table 3.

We then estimated impacts of projected VPD on maize yield for three hybrid age groups: old (2000–2006), intermediate (2007–2013), and newer (2016–2020) hybrids as follows:

$$imp_{VPD,g} = \frac{\sum_{p=1}^{2} (\beta_{3,p} \times VPD_{b,p} + \beta_{4,g,p} \times VPD_{a,p})}{\hat{Y}} \times 100 \qquad (10)$$

where $imp_{VPD}$ refers to projected impacts of VPD on yields (%) relative to historical averaged yields ($\hat{Y}$; 12.5 t ha$^{-1}$, Fig. 1c) and *g* refers to either old, intermediate, or new hybrids.

### Reporting summary

Further information on research design is available in the Nature Portfolio Reporting Summary linked to this article.

## Data availability

The raw maize hybrid data used in this study are publicly available at the Figshare repository: https://doi.org/10.6084/m9.figshare.28960106. Historical weather data from the Integrated Surface Dataset are publicly available at https://www.ncei.noaa.gov/access/metadata/landing-page/bin/iso?id=gov.noaa.ncdc:C00532. CMIP6 data are publicly available at https://www.nccs.nasa.gov/services/data-collections/land-based-products/nex-gddp-cmip6. Source data are provided with this paper.

## Code availability

Data processing, statistical analysis, and figure generation were performed using MATLAB 2023a. The code supporting main findings of this study is publicly available at https://doi.org/10.6084/m9.figshare.27038308.

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

## Acknowledgements

This study was supported by the U.S. National Science Foundation (nos. 2420405 and FAIN:2345039) (X.L.) and USDA Agricultural Research Service (A22-0103-001) (X.L.). We thank Steve Watson for his contribution in editing and finalizing the paper. Contribution no. 25-033-J from the Kansas Agricultural Experiment Station.

## Author contributions

H.Z. and X.L. designed and coordinated the research and conducted the analysis. J.B.T., G.J.K., M.B.K., and G.F.S. interpreted the results and advised on presentation of the main findings. H.Z. and L.Z. collected all the needed data. H.Z. and X.L. wrote the manuscript. N.W., Z.L., J.Z., A.A., P.H.G. and other coauthors revised the manuscript.

## Competing interests

The authors declare no competing interests.
