## [Peer Review file · Nature Communications]

Concurrent improvements in maize yield and drought resistance through breeding advances in the U.S. Corn Belt

Corresponding Author: Professor Xiaomao Lin

Version 0:

Reviewer comments:

Reviewer #1

(Remarks to the Author)

Below I provide first some general comments and my impressions of the manuscript.

Following the general comments, I provide some specific comments related to specific sections of the manuscript that the authors could consider revising to improve their manuscript.

General Comments

For a comprehensive maize multi-environment trial data set for the US corn-belt the authors apply combinations of environmental Vapor Pressure Deficit (VPD) and precipitation covariates, broken out by two crop growth stages (VEG and GFP) and an estimate of environmental stress level based on mean grain yield of all tested hybrids (three levels of an environment index) to characterise hybrid yield performance differences. Their objective is to evaluate the realised improvements of hybrid performance for yield and “resistance to drought” or “drought sensitivity” for contrasts between groups of hybrids designated as drought tolerant and not drought tolerant. Their analysis is comprehensive and convincing.

The authors note “To track yield progress under varying environmental stress levels, a precise measure of stress experienced by crops is key. This measure is often quantified as average yield across all hybrids grown at a given site-year, known as the environment index (EI)^{37,38}.” At best the chosen “environment index” is a coarse-grained measure of environment stress level. The authors should qualify their comments in relation to their chosen environmental index. I have made some other cautionary comments below in the Specific Comments section. The authors find that in their data set “This suggests that recent breeding progress has consistently increased maize yield under both favorable and stressed environmental conditions.” This is an important result found in a comprehensive independent data set. Comparable results have been reported elsewhere, but in a much smaller data set. The authors then focus on VPD as a specific component of atmospheric stress for further analysis. They demonstrate in their MET data set that the group of “drought tolerant” hybrids perform better than the “not drought tolerant” group of hybrids. Their results align with the recent results reported by Messina et al. (2023).

Messina CD, Gho C, Hammer GL, Tang T, Cooper M (2023) Two decades of harnessing standing genetic variation for physiological traits to improve drought tolerance in maize. *Journal of Experimental Botany* 74:4847-4861

The results reported by the authors are an informative demonstration of the realised impact of long-term breeding program efforts. Importantly the authors demonstrate the important contribution of breeding for improved drought tolerance of maize in a comprehensive MET representing the Target Population of Environments of the maize breeding programs. The data set is comprehensive, and their chosen analysis approach is effective in demonstrating important realised genetic gain for yield and drought tolerance of maize achieved by targeted long-term breeding.

Specific Comments

Abstract

Lines 32-34: “Recent breeding advances are recognized as a strategy to enhance yield and drought resistance, yet the extent of their synchronous improvements has not been systematically documented.” This statement is incorrect, and the authors should revise to reflect other relevant studies reported in the literature.

Below are four relevant publications that investigate the synchronous improvement of yield and drought resistance for maize in the US corn-belt that the authors have not consulted or at least not cited in their manuscript. The authors need to qualify their statement. Their study provides a large-scale evaluation of the realised impact of accelerated breeding for both yield and drought resistance in the US corn-belt, as has been the target of industry breeding programs over the last two decades.

The four manuscripts listed below provide more background context to this accelerated breeding effort related to the AQUAmax hybrids that are an important and major component of their reported MET data set.

Cooper M, Tang T, Gho C, Hart T, Hammer G, Messina C (2020) Integrating genetic gain and gap analysis to predict improvements in crop productivity. *Crop Science* 60:582-604

Messina CD, Gho C, Hammer GL, Tang T, Cooper M (2023) Two decades of harnessing standing genetic variation for physiological traits to improve drought tolerance in maize. *Journal of Experimental Botany* 74:4847-4861

Cooper M, Messina CD (2023) Breeding crops for drought-affected environments and improved climate resilience. *The Plant Cell* 35:162-186

Cooper M, Messina CD, Tang T, Gho C, Powell OM, Podlich DW, Technow F, Hammer GL (2023) Predicting genotype x environment x management (GxExM) interactions for design of crop improvement strategies: integrating breeder, agronomist, and farmer perspectives. *Plant Breeding Reviews* 46:467-585

Lines 34-35: "compiling a dataset of 92,096 maize field trials" Lines 78-79 state "92,096 hybrid-trial observations across 12,847 hybrids". Lines 265-266: "In total, the database we used included 92,096 data points across 63 field sites." It appears that the 92,096 are field plot observations, not 92,096 field trials as indicated in the abstract. The authors should clarify the actual number of field trials that comprise their data set if this is what they want to report in the abstract. Alternatively, the authors could indicate that they have a data set with 92,096 yield data records.

Lines 63-65: "In response to these challenges, breeders have worked over the past two decades to improve drought resistance in maize through the incorporation of drought-resistance traits into genetically modified (GM) trait packages^{17,20,21}" It is not clear whether the authors are proposing that recent advances have only focused and relied on application of transgenic methods? The majority of the hybrids with improved performance under drought conditions from the last 2 decades were developed using improved accelerated breeding achieved by combining advanced phenotyping and genomic prediction, rather than relying on transgenic modifications for drought tolerance. The authors should clarify and correct their statement as required.

Lines 65-66: "After the widespread drought in 2012, which caused a 30% yield loss, breeders intensified the introduction of drought tolerance (DT) traits." The relevant breeding work had actually already commenced in the decade before the 2012 drought. The DuPont-Pioneer AQUAmax maize hybrids were first launched in 2011 for US farmers. The rapid adoption of the improved hybrids by US farmers followed the 2012 drought when their superior performance was demonstrated across the US corn-belt in 2012 as was documented by the cited publication by Gaffney et al. (2015). Further improvements beyond the first generation of drought tolerant hybrids was demonstrated in the publication by Messina et al. (2023).

Messina CD, Gho C, Hammer GL, Tang T, Cooper M (2023) Two decades of harnessing standing genetic variation for physiological traits to improve drought tolerance in maize. *Journal of Experimental Botany* 74:4847-4861

Lines 69-71: "However, the speed and effectiveness of breeding improvements to drought resistance across diverse environments, and the underlying mechanisms driving these changes, remain poorly understood. Addressing this knowledge gap is critical for targeting future breeding strategies and providing information for research investment policies that aim to ensure sustainable food production in the future^{25,26}" As indicated above the authors have missed some key literature that has documented the key underlying mechanisms that were targeted in breeding the drought tolerant hybrids. They should consult the relevant literature indicated and revise their statement accordingly.

Lines 299-300: "providing a reliable measure of stress level in field trials". The authors, as have many others, assume that the mean yield of all hybrids included in a trial provides a measure "stress level". The use of this environmental index provides a coarse-grained relative measure of trial yield level. It has many limitations for measuring "stress level". The same environmental mean yield level can be achieved through many different yield-reducing mechanisms, including poor establishment, low plant population, disease, lodging, high temperature, and many others. The authors should at least connect their choice of environmental mean yield as an index to the large body of literature on the advantages and disadvantages of such an approach.

(Remarks on code availability)

The provided code was only to generate figures.

This in itself was not central to the main analyse conducted in the manuscript, but for visualisation of results.

The data sources used are publicly available and I consider that the linear models that the authors report can easily be implemented using a number of available software tools for mixed model analysis.

A major undertaking by the authors has been the processing of the publicly available data sets to generate the appropriate factors that were used to conduct the chosen linear models.

Reviewer #2

(Remarks to the Author)

Comments on "Parallel gains in maize yield and drought resistance driven by breeding advances in the U.S. Corn Belt"

This research focus on a critical question about yield gain differential between drought tolerant maize, and the role of the transpiration response to vapor pressure deficit (vpd) at a regional scale. While predicting the future is difficult, it is anticipated that at least atmospheric drought will increase with increasing anthropogenic climate change. Water deficits can compromise further the capacity of the global food system to provide calories and animal proteins to a growing population.

The research presented in this manuscript is relevant to science and society.

To provide an approximate answer to the question above, Zhao et al. used a statistical approach applied to a very large data comprised of about 20 years of multi-environment trials conducted in the U.S. corn belt. The data included commercial genotypes that were proven to have higher levels of tolerance to drought relative to experimental controls.

The main findings could be summarized as 1) a breakpoint in the relationship between yield and vpd exist at 1.3-1.4 kPa, 2) yield decrease with increasing vpd above the vpd threshold, 4) drought tolerance in maize is largely explained by genetic improvement during the grain filling period, and 3) a non-linear association between yield and precipitation.

While the values reported by Zhao et al. on the vpd breakpoint are lower than most publications by Sinclair et al. they conform with hundreds of measurements conducted by this reviewer (unpublished). The negative response of yield to vpd above the vpd threshold conforms with the physiological basis of yield determination in maize. A reduction in stomatal conductance due to vpd restricts carbon assimilation and therefore growth and yield. This is magnified during grain filling as most carbon stem from current photosynthesis. The results conform with theoretical predictions, therefore contributing to the advancement of scientific inquiry. Finally, results conform well with physiological understanding of genetic improvement for drought tolerance in maize, mainly attributed to gains during the reproductive period (grain filling in this manuscript).

It is important to note that the findings by Zhao et al. are consistent with theory applied to the central and eastern U.S. corn belt where soil water deficits are usually not severe. A restriction in transpiration, and thus a conservation of water, is not a useful mechanism for corn to deal with water deficits. Often, water conservation is negatively related to yield. The relationship that Zhao et al. found were tailored to the data used that covered most of the central and eastern U.S. corn belt. This is important to note, because in drier environments such as in the western U.S. corn belt including parts of Nebraska, Kansas, and Texas, water conservation strategies can and usually underpin drought tolerance in maize. If future climates are drier than current climates, it will be important to have models that can predict both responses. I suggest for the authors to include data from multi-environment trials conducted by land grant universities in the western states. The authors should find the opposite result as those reported in the current manuscript, increasing the applicability of the models and strengthening the model itself.

Developing a model that has broad applicability is necessary, yet not sufficient, to make predictions about impacts of climate change. As I indicate above, what the authors found as a negative yield response to water deficit in the central corn belt may turn out to be a positive relation when water deficits occur during grain filling, as one may expect in warmer and drier climates. I don't think the speculation about impacts of climate change is appropriate.

Because crop improvement is dynamic it is not sufficient to assess the impacts of climate change using a static model. While transpiration (and yield) response to vpd underpin drought tolerance in contemporary germplasm, other mechanisms may underpin drought tolerance in future cohorts of hybrids. Breeders select in a mixture of environments and test in a set of environments that is correlated to some degree. The degree and sign of the correlation dictates if how quickly the breeding programs can adapt to the changing climate. A recent study shows how Corteva breeding adapted the germplasm despite the changing climate (<https://www.biorxiv.org/content/10.1101/2023.09.19.558447v1>). During this period many physiological traits changed underpinning the adaptation of maize to drier conditions in the western corn belt and wetter conditions in the central corn belt. This section of the manuscript needs to be revised.

A minor comment relates to the effect of CO₂ fertilization in maize. Due to the C₄ photosynthesis pathway, it is unlikely that maize yields will respond to CO₂ directly but through an improved water use efficiency.

(Remarks on code availability)

Reviewer #3

(Remarks to the Author)

Zhao et al. described a comprehensive study to analyze the yield trials of U.S. Corn Belt and environmental data to draw a conclusion about the genetic improvement of corn at a large scale. Authors carefully collected available data from 3 key corn production states and conducted systematic analyses. At the first read, it did not appear to be anything special other than a large data set. However, upon reflection, I realized that this was indeed a great way to identify patterns from the data and there was no other feasible way to run new experiments to test this. Any designed experiments would not be able to capture the actual hybrid-environment combinations happened in the past at such an extensive trial scale, including that earlier era hybrid study from one major company, the ongoing era hybrid study from a different company, or the ongoing G2F initiative. The analysis procedure makes sense and the conclusion is very justified. This study has significant impact on setting the record straight than a previous small-scale study (marginal corn production area), and correctly reflect what has been going on in crop improvement and production. I commend authors for working on this important research area.

I have the following comments for authors consideration.

1) It would be helpful to have some generic diagrams as supplementary figures to support the main figures and to show the several important concepts in the study. Text description does not always do a good job. “the release year-specific average yield”, how “vegetative” and “grain-filling period” were identified for hybrid/site-year, how the compositions of three groups of hybrids change along the year (2000-2020) since they only appeared in some years, how often DT hybrids landed in one of the three groups, and the existence of LYH-MYH-HYH, the old-median-new, and nonDT-DT asks for any further sub-setting if possible (?).

2. Fig. S5 did not seem to provide clear and adequate justification for the selection of these two thresholds in Fig. 2. I was expecting more evidence as this part is very critical to the analysis of this comprehensive dataset.

3. Fig. 3a-c and associated analyses and results need more attention. You did two way grouping based on the marginal means, site-year mean and hybrid mean, and generated the plots. Not sure why the test was conducted. And if $p > 0.05$, does not mean there is no enough evidence for your separation. Also, check you statistics, “Bars refer to the 576 mean trend, and error bars are 95% confidence interval”. The standard error for the mean is generally very small with large sample size, and the 95% confidence interval for the mean would not be that wide.

4. Fig. 3d-f, please indicate the difference between the red (base lines with the fixed slope 1.0s, each group of hybrids by themselves (check?)) and the rest. What was plotted and the legend text, “Dashed lines show the fitted linear regression for each environmental index level” are not easy to understand, since you introduced the year here. You probably do not have to do a comprehensive analysis to declare the significant difference among those slopes. But I hope some of the slopes are different from others.

5. Fig. 4a, any reasons to explain the wild swings of the sensitivity? I think authors know the documented rainfall and temperature patters that led to this. Any need to calculate the slopes for the shaded areas? Not sure whether some un-adapted hybrids from some companies outside of U.S. were removed from the analysis. Also, please correct this statement, “We found a slight decrease ($P > 0.10$) in maize yield resistance to drought during the VEG period,” You cannot have both ways. Not significant, so no change.

6. Fig. 4a, would it be good to use the natural production average to partition the data into two groups, those “normal years” to obtain the slope and those above and below the trend line years. But I suspect it would cancel out? Even this is the case, feels like establishing some sort of connection with the production would be beneficial to this study.

7. Figs. 4c and the text. This VPD increase was for what stage, GFP?

8. Not criticism. The compiled data needs to be made available in the final stage. Indicating the source is not adequate for checking and re-analysis.

(Remarks on code availability)

Version 1:

Reviewer comments:

Reviewer #1

(Remarks to the Author)

The authors have provided a comprehensive response to the comments, suggestions and questions raised by the three reviewers.

I commend the authors on providing a well-structured and informative response document that explains the improvements made to their manuscript.

One final aspect that the authors could include to further improve their manuscript is clarify the importance of long-term selection for integrated Hybrid-by-Management technology combinations for the long-term maize productivity gains and reduction in hybrid sensitivities to environmental conditions. This has been well documented by Duvick in his many studies of long-term genetic improvement of hybrids from commercial breeding targeted at the US corn-belt.

While the authors' data set does not provide a structured sequence of experiments across years with a factorial of hybrids and agronomic management strategies (such as the experiments conducted by Duvick and colleagues), the authors have done some inciteful breakouts of good, normal and bad years (Fig. R3.13) in response to a suggestion by Reviewer 3. This breakout revealed evidence of an improvement (reduced sensitivity) to VPD in bad years for the GFP. This is an important result. The authors can connect this finding from their analyses with the results reported as Figure 8.21 in the cited publication by Cooper, M. et al. Predicting Genotype \times Environment \times Management ($G \times E \times M$) interactions for the design of crop improvement strategies: integrating breeder, agronomist, and farmer perspectives. Plant Breeding Reviews 46, 467-585 (2022). In Figure 8.21 within Cooper et al. (2022), and the associated text, these authors demonstrate the sequence of improvements in hybrid yield performance under drought conditions, achieved by shifting hybrid water use patterns by breeding, resulted in reduced the sensitivities of hybrids to droughts during flowering time and grain filling periods. This allowed farmers to shift their agronomic management to higher plant densities in combination with changes in irrigation management (for the Western regions) to further protect hybrids during the grain filling period. The trends in improvements in GFP performance of hybrids under stress that the authors have revealed, their Fig. R3.13 from re-analysis of their data sets, are consistent with the shifts in hybrid-by-management trends that were reported by Cooper et al. in their Fig. 8.21.

(Remarks on code availability)
No additional review of code.

Reviewer #2

(Remarks to the Author)

I praise the authors for considering my suggestions. Incorporating prior literature made this paper complete. Including data from the far west corn belt make this paper much stronger than the prior version. The additional analysis that led to the following conclusions: 1) no changes in VEG but GF, 2) response to VPD underpins DT, 3) there is parallel gain (this was deliberate in AQUAmax breeding objectives), and 4) the percent change between DT and non-DT estimated in this study, are very consistent with what we know about the physiological changes underpinning drought tolerance and provide an independent verification of other studies are cited in the manuscript.

(Remarks on code availability)

Reviewer #3

(Remarks to the Author)

After going through the entire response and the revised manuscript carefully, I conclude that authors made genuine efforts to address concerns raised by the reviewers. I am satisfied with the explanation in the response and modification in the revised manuscript. This is a very well conceived and completed study, and a significant contribution to the literature!

A few minor comments:

1. Given the comprehensiveness of this study, it is very helpful to document many detailed analysis and decision steps authors used. Without that, it can even be challenging for authors (maybe except 1st and last author) to repeat all the steps. So, I encourage authors to go through the response document to see whether you can further incorporate any text of reasoning and decision-making text into the supplementary materials. There is also some space (limited in 5,000 words; 3,305) in the main text to accommodate some critical additions. This can be mainly achieved with expanded supplementary texts and supplementary figure captions, if not supplementary methods. This is a generic request to check, not specific parts, so that authors can decide what is best for their paper.
2. Abstract, Results (L207-227), Figure 5, and related supplementary materials. Please do a targeted reading through to make sure the terms you want to use: old, intermediate, and new; new and new age; and advanced and older.
3. Please think about the title "parallel gains in maize yields and drought tolerance". Did you want to stress 1) the "parallel" between maize yield and drought tolerance, or 2) "parallel" among different yield groups of hybrids, different environmental conditions, different age groups of hybrids? Good to look at the main messages supported by the main figures and then Abstract to construct a title that is easy to grasp. These 1) and 2) can be somewhat related, but not the same since additional reasoning is needed when you stress in the Abstract, "these gains are accompanied by enhanced drought resistance during the grain filling period", after "provides evidence of consistent yield increases across diverse environmental conditions."

Others:

- L631. Modify "The calculated process".
- L638. Rewrite text for panel b.
- L661. It may be better to omit "(SSP)" here.

(Remarks on code availability)

Point-by-point responses to three Referees

The authors have used regular fonts for the Referees' comments (which might be divided into two or multiple comments), blue fonts for our responses, and red fonts with quotation marks to show the revised text. In this point-by-point-response, the figures are numbered as follows. The R stands for "Response" and the numbers refer to the Referees and their comments. For example, Fig. R1.2 means that the figure is in the "Responses" and it is the 2nd figure in response to Referee 1.

Referee 1

Reviewer #1 (Remarks to the Author)

Below I provide first some general comments and my impressions of the manuscript. Following the general comments, I provide some specific comments related to specific sections of the manuscript that the authors could consider revising to improve their manuscript.

General Comments

For a comprehensive maize multi-environment trial data set for the US corn-belt the authors apply combinations of environmental Vapor Pressure Deficit (VPD) and precipitation covariates, broken out by two crop growth stages (VEG and GFP) and an estimate of environmental stress level based on mean grain yield of all tested hybrids (three levels of an environment index) to characterise hybrid yield performance differences. Their objective is to evaluate the realised improvements of hybrid performance for yield and "resistance to drought" or "drought sensitivity" for contrasts between groups of hybrids designated as drought tolerant and not drought tolerant. Their analysis is comprehensive and convincing.

Response: Thank you for your complimentary and encouraging comments. Your review and a few key papers that you kindly brought to us substantially improved our paper quality. We appreciate your thorough and insightful suggestions. Our point-by-point response to all your comments are below.

The authors note "To track yield progress under varying environmental stress levels, a precise measure of stress experienced by crops is key. This measure is often quantified as average yield across all hybrids grown at a given site-year, known as the environment index (EI)^{37,38}." At best the chosen "environment index" is a coarse-grained measure of environment stress level. The authors should qualify their comments in relation to their chosen environmental index. I have made some other cautionary comments below in the Specific Comments section.

Response: We agree that "environment index is a coarse-grained measure of environment stress level," as it provides an overall grading of environmental conditions by integrating factors such as soil properties, management practices, and climate variability.

The environment index (EI), originally proposed by Yates and Cochran (1938) and later formalized by Finlay and Wilkinson (1963), is defined as the mean yield of all varieties for each site-year. This approach has been widely used in multilocation crop variety trials to assess the adaptation of breeding programs to varied environmental conditions. For instance, Xiong et al. (2021) employed

EI to evaluate the adaptation of global wheat breeding to climate change, highlighting its utility in analyzing large-scale trends in crop performance.

In our study, we used the EI to assess whether breeding advancement over the past two decades consistently improved yield gains across a wide range of realized environmental conditions. By stratifying environmental conditions into five levels from low-yielding (stressful) to high-yielding (favorable) environments based on EI values, we were able to assess the overall trends in breeding progress under varying environmental stress levels - a practical means to assess breeding performance across a spectrum of environmental conditions that accommodate the inherent complexity of natural field conditions.

As an example, to illustrate EI here, we analyzed the average accumulated precipitation during the maize growing season for EI levels (**Fig. R1.1**). The results showed a distinct pattern: the highest EI (lowest stress) corresponded to the highest precipitation amounts (540 mm) for corn growing and production, which was 102 mm higher than that observed for the lowest EI (highest stress).

Fig. R1.1. Accumulated precipitation during maize growing season for five levels of the environment index from the highest environmental index level (red color) representing the best yield conditions, to the lowest environmental index level (blue color) indicating the worst yield conditions. Bars refer to the mean precipitation, and error bars are standard errors.

However, EI is an aggregate measure of overall environmental conditions in the field and does not distinguish specific sources of stress, such as drought, disease and pest pressures, poor establishment, and lodging. Based on your detailed suggestion in “specific comments,” we revised two sentences in Method section to better reflect the context and limitations of the environment index, as follows:

Lines 333-339: “...providing an overall environmental level in field trials. This approach, widely used in plant breeding, enables the assessment of crop variety/hybrids adaptation across diverse environmental conditions by offering a simplified yet integrative measure of complex natural

environments. Although this approach does not pinpoint specific environmental stressors or yield-reducing mechanisms, such as poor establishment, pest and disease, lodging, or abiotic stresses, it remains a practical tool for capturing patterns of crop performance across varying levels of integrated environmental stress.”

References:

- Yates, F., & Cochran, W. G. (1938). The analysis of groups of experiments. *J. Agric. Sci.* 28(4), 556-580.
- Finlay, K. W., & Wilkinson, G. N. (1963). The analysis of adaptation in a plant-breeding programme. *Aust. J. Agric. Res.* 14(6), 742-754.
- Xiong, W., Reynolds, M. P., Crossa, J., Schulthess, U., Sonder, K., Montes, C., ... & Payne, T. (2021). Increased ranking change in wheat breeding under climate change. *Nat. Plants* 7(9), 1207-1212.

The authors find that in their data set “This suggests that recent breeding progress has consistently increased maize yield under both favorable and stressed environmental conditions.” This is an important result found in a comprehensive independent data set. Comparable results have been reported elsewhere, but in a much smaller data set. The authors then focus on VPD as a specific component of atmospheric stress for further analysis. They demonstrate in their MET data set that the group of “drought tolerant” hybrids perform better than the “not drought tolerant” group of hybrids. Their results align with the recent results reported by Messina et al. (2023).

Messina CD, Gho C, Hammer GL, Tang T, Cooper M (2023) Two decades of harnessing standing genetic variation for physiological traits to improve drought tolerance in maize. *Journal of Experimental Botany* 74:4847-4861 The results reported by the authors are an informative demonstration of the realised impact of long-term breeding program efforts. Importantly the authors demonstrate the important contribution of breeding for improved drought tolerance of maize in a comprehensive MET representing the Target Population of Environments of the maize breeding programs. The data set is comprehensive, and their chosen analysis approach is effective in demonstrating important realised genetic gain for yield and drought tolerance of maize achieved by targeted long-term breeding.

Response: We appreciate your detailed and encouraging comments. We agree that the study by Messina et al. (2023) provides a compelling demonstration of genetic gains in maize drought tolerance. We have benefited from the insights in their work, particularly in terms of connecting genotype improvement to managed stress environments (MSEs).

Specific Comments

Abstract

Lines 32-34: “Recent breeding advances are recognized as a strategy to enhance yield and drought resistance, yet the extent of their synchronous improvements has not been systematically documented.” This statement is incorrect, and the authors should revise to reflect other relevant studies reported in the literature. Below are four relevant publications that investigate the synchronous improvement of yield and drought resistance for maize in the US corn-belt that the authors have not consulted or at least not cited in their manuscript. The authors need to qualify their statement. Their study provides a large-scale evaluation of the realised impact of accelerated breeding for both yield and drought resistance in the US corn-belt, as has been the target of industry breeding programs over the last two decades.

The four manuscripts listed below provide more background context to this accelerated breeding effort related to the AQUAmax hybrids that are an important and major component of their reported MET data set.

Cooper M, Tang T, Gho C, Hart T, Hammer G, Messina C (2020) Integrating genetic gain and gap analysis to predict improvements in crop productivity. *Crop Science* 60:582-604

Messina CD, Gho C, Hammer GL, Tang T, Cooper M (2023) Two decades of harnessing standing genetic variation for physiological traits to improve drought tolerance in maize. *Journal of Experimental Botany* 74:4847-4861

Cooper M, Messina CD (2023) Breeding crops for drought-affected environments and improved climate resilience. *The Plant Cell* 35:162-186

Cooper M, Messina CD, Tang T, Gho C, Powell OM, Podlich DW, Technow F, Hammer GL (2023) Predicting genotype x environment x management (GxExM) interactions for design of crop improvement strategies: integrating breeder, agronomist, and farmer perspectives. *Plant Breeding Reviews* 46:467-585

Response: Thank you for highlighting these important studies. These papers provide thorough and invaluable insights into maize breeding efforts. They investigated maize breeding progress for improved drought tolerance under water-limited and/or water-sufficient conditions based on environments (Messina et al., 2023; Cooper and Messina, 2023). They proposed a biophysical framework using the Crop Growth Model (CGM) to advance G×E×M prediction (30% prediction accuracy improvement) and established that drought breeding closed the genetic gain gap, with improvement under drought ($7.5 \text{ g m}^{-2} \text{ yr}^{-1}$) to non-stress ($8.6 \text{ g m}^{-2} \text{ yr}^{-1}$) over the last two decades (Cooper et al., 2020; Cooper and Messina, 2023; and Cooper et al., 2023).

Based on your excellent suggestion, we revised the sentence in the abstract:

Lines 32-35: “Recent studies have examined the genetic and physiological bases of yield and drought tolerance improvements in maize; however, comprehensive, field-based quantification of synchronous improvements of yield and drought resistance across diverse environmental conditions remain limited.”

Additionally, we modified one sentence in the Introduction section to acknowledge prior studies on maize yield gains and drought resilience over breeding and cited these relevant sources. This modification was merged into your comments in “Lines 69-71:”

Lines 34-35: “compiling a dataset of 92,096 maize field trials” Lines 78-79 state “92,096 hybrid-trial observations across 12,847 hybrids”. Lines 265-266: “In total, the database we used included 92,096 data points across 63 field sites.” It appears that the 92,096 are field plot observations, not 92,096 field trials as indicated in the abstract. The authors should clarify the actual number of field trials that comprise their data set if this is what they want to report in the

abstract. Alternatively, the authors could indicate that they have a data set with 92,096 yield data records.

Response: Thank you. The number “92,096” refers to hybrid-trial observations rather than field trials. We modified “compiling a dataset of 92,096 maize field trials”, as follows:

Line 36: “...92,096 hybrid-trial observations...”

Lines 63-65: “In response to these challenges, breeders have worked over the past two decades to improve drought resistance in maize through the incorporation of drought-resistance traits into genetically modified (GM) trait packages^{17,20,21}” It is not clear whether the authors are proposing that recent advances have only focused and relied on application of transgenic methods? The majority of the hybrids with improved performance under drought conditions from the last 2 decades were developed using improved accelerated breeding achieved by combining advanced phenotyping and genomic prediction, rather than relying on transgenic modifications for drought tolerance. The authors should clarify and correct their statement as required.

Response: Thank you for your insightful comment. We appreciate the opportunity to clarify this point. You are correct that the development of most commercial drought-tolerant maize hybrids over the past two decades has relied on improved accelerated breeding achieved by combined advanced phenotyping and genomic prediction (Cooper et al., 2014; Messina et al., 2023).

We modified the sentence to reflect this broad context and added the two relevant references (Cooper et al., 2014; Messina et al., 2023), as follows:

Lines 64-66: “..., breeders have enhanced drought resistance traits in maize over the past two decades primarily through improved accelerated breeding achieved by combining advanced phenotyping and genomic prediction^{17,20-22}.”

References:

Cooper, M., Gho, C., Leafgren, R., Tang, T., & Messina, C. (2014). Breeding drought-tolerant maize hybrids for the US corn-belt: discovery to product. *J. Exp. B.* 65(21), 6191–6204.

Messina, C. D., Gho, C., Hammer, G. L., Tang, T., & Cooper, M. (2023) Two decades of harnessing standing genetic variation for physiological traits to improve drought tolerance in maize. *J. Exp. B.* 74(16), 4847-4861.

Lines 65-66: “After the widespread drought in 2012, which caused a 30% yield loss, breeders intensified the introduction of drought tolerance (DT) traits.” The relevant breeding work had actually already commenced in the decade before the 2012 drought. The DuPont-Pioneer AQUAmax maize hybrids were first launched in 2011 for US farmers. The rapid adoption of the improved hybrids by US farmers followed the 2012 drought when their superior performance was demonstrated across the US corn-belt in 2012 as was documented by the cited publication by Gaffney et al. (2015). Further improvements beyond the first generation of drought tolerant hybrids was demonstrated in the publication by Messina et al. (2023).

Messina CD, Gho C, Hammer GL, Tang T, Cooper M (2023) Two decades of harnessing standing genetic variation for physiological traits to improve drought tolerance in maize. *Journal of Experimental Botany* 74:4847-4861

Response: Thank you for this important context and providing us with excellent studies (Gaffney et al., 2015; Messina et al., 2023) which thoroughly document the development of drought tolerance hybrids. Indeed, dedicated research efforts in the early 2000s emerged and resulted in the commercialization of AQUAmax® hybrids in 2011 (Messina et al., 2023).

Based on your suggestion, we rewrote the following sentence,

Lines 66-68: “which caused a 30% yield loss, the adoption of drought-tolerant (DT) maize hybrids by U.S. farmers substantially increased, with over 20% of the total area planted with DT hybrids in 2016.”

Lines 69-71: “However, the speed and effectiveness of breeding improvements to drought resistance across diverse environments, and the underlying mechanisms driving these changes, remain poorly understood. Addressing this knowledge gap is critical for targeting future breeding strategies and providing information for research investment policies that aim to ensure sustainable food production in the future^{25,26}” As indicated above the authors have missed some key literature that has documented the key underlying mechanisms that were targeted in breeding the drought tolerant hybrids. They should consult the relevant literature indicated and revise their statement accordingly.

Response: Thank you. We incorporated the suggested references and revised the text as follows:

Lines 70-77: “Furthermore, recent studies have shown that dedicated breeding and technology advancements over the past two decades have markedly enhanced the genetic gains of maize yield under water-limited conditions, increasing from 0.06 t ha⁻¹ year⁻¹ to 0.08 t ha⁻¹ year⁻¹^{22,26-28}. This progress has mainly been achieved by the selection of multiple drought-tolerance mechanisms, such as shifting the patterns of water use. While these findings are well-documented in field studies, the speed and effectiveness of breeding improvements to drought resistance across diverse environments in large-scale maize breeding trials remain poorly understood.”

Lines 299-300: “providing a reliable measure of stress level in field trials”. The authors, as have many others, assume that the mean yield of all hybrids included in a trial provides a measure “stress level”. The use of this environmental index provides a coarse-grained relative measure of trial yield level. It has many limitations for measuring “stress level”. The same environmental mean yield level can be achieved through many different yield-reducing mechanisms, including poor establishment, low plant population, disease, lodging, high temperature, and many others. The authors should at least connect their choice of environmental mean yield as an index to the large body of literature on the advantages and disadvantages of such an approach.

Response: Thank you for your insight and suggestions on the environment index. Please see our previous response in your general comments.

(Remarks on code availability)

The provided code was only to generate figures.

This in itself was not central to the main analyse conducted in the manuscript, but for visualisation of results.

Response: Thank you. We have uploaded three codes for main analysis including 1) estimating impact of vapor pressure deficit (VPD) on maize yield (Fig. 2); 2) calculating yield trends at different levels of environmental index (Fig. 3); and 3) estimating sensitivity of maize yield to VPD for each first trial-year hybrids (Fig. 4). These are the main results for this manuscript.

The data sources used are publicly available and I consider that the linear models that the authors report can easily be implemented using a number of available software tools for mixed model analysis.

A major undertaking by the authors has been the processing of the publicly available data sets to generate the appropriate factors that were used to conduct the chosen linear models.

Response: Thank you for your understanding. We have uploaded source data for each figure in the main manuscript. We will also provide the original data at the final stage.

-End of Response for Referee 1.

Reviewer #2 (Remarks to the Author)

Comments on “Parallel gains in maize yield and drought resistance driven by breeding advances in the U.S. Corn Belt”

This research focus on a critical question about yield gain differential between drought tolerant maize, and the role of the transpiration response to vapor pressure deficit (vpd) at a regional scale. While predicting the future is difficult, it is anticipated that at least atmospheric drought will increase with increasing anthropogenic climate change. Water deficits can compromise further the capacity of the global food system to provide calories and animal proteins to a growing population. The research presented in this manuscript is relevant to science and society.

To provide an approximate answer to the question above, Zhao et al. used a statistical approach applied to a very large data comprised of about 20 years of multienvironment trials conducted in the U.S. corn belt. The data included commercial genotypes that were proven to have higher levels of tolerance to drought relative to experimental controls.

The main findings could be summarized as 1) a breakpoint in the relationship between yield and vpd exist at 1.3-1.4 kPa, 2) yield decrease with increasing vpd above the vpd threshold, 4) drought tolerance in maize is largely explained by genetic improvement during the grain filling period, and 3) a non-linear association between yield and precipitation.

While the values reported by Zhao et al. on the vpd breakpoint are lower than most publications by Sinclair et al. they conform with hundreds of measurements conducted by this reviewer (unpublished). The negative response of yield to vpd above the vpd threshold conforms with the physiological basis of yield determination in maize. A reduction in stomatal conductance due to vpd restricts carbon assimilation and therefore growth and yield. This is magnified during grain filling as most carbon stem from current photosynthesis. The results conform with theoretical predictions, therefore contributing to the advancement of scientific inquiry. Finally, results conform well with physiological understanding of genetic improvement for drought tolerance in maize, mainly attributed to gains during the reproductive period (grain filling in this manuscript).

Response: Thank you for your complimentary and encouraging comments. Our point-by-point response to all your comments are below.

It is important to note that the findings by Zhao et al. are consistent with theory applied to the central and eastern U.S. corn belt where soil water deficits are usually not severe. A restriction in transpiration, and thus a conservation of water, is not a useful mechanism for corn to deal with water deficits. Often, water conservation is negatively related to yield.

Response: Thank you for acknowledging that our findings are consistent with theoretical expectations for the central and eastern U.S. corn belt.

The relationship that Zhao et al. found were tailored to the data used that covered most of the central and eastern U.S. corn belt. This is important to note, because in drier environments such as in the western U.S. corn belt including parts of Nebraska, Kansas, and Texas, water

conservation strategies can and usually underpin drought tolerance in maize. If future climates are drier than current climates, it will be important to have models that can predict both responses. I suggest for the authors to include data from multi-environment trials conducted by land grant universities in the western states. The authors should find the opposite result as those reported in the current manuscript, increasing the applicability of the models and strengthening the model itself.

Response: Thank you for your thoughtful insights and suggestions to incorporate data from the western U.S. corn belt. We agree that expanding the dataset can enhance the generalizability of our findings and have conducted additional analyses by combining data from the western U.S. corn belt. However, based on further analysis, we found that the response of maize yield to high VPD (VPD above the threshold) remained negative when integrating western and central data, which was consistent with our findings in the main manuscript.

Based on your suggestions, a limited transpiration (LT) trait offers generally greater yield gains relative to a non-limited transpiration (non-LT) trait under drought-prone regions, such as those in the western U.S. corn belt, by increasing the partitioning of water use from the vegetative to reproductive stage of crop development (Cooper et al., 2014; Messina et al., 2015). A classical case from Messina et al. (2015) has greatly contributed to our understanding of LT gains across the U.S. corn belt. Using a process-based crop model, they identified that a positive yield difference between the LT trait and non-LT trait was more noticeable in lower-yield environments, such as the western region, while a slightly negative yield difference was observed in higher-yield environments (**Fig. R2.1a**). This finding highlights the role of the LT trait in enhancing drought tolerance across different regions.

Fig. R2.1. a, Change in yield difference between limited transpiration (LT) and non-limited transpiration (non-LT) over yield conditions in non-LT traits. Points represent median value, and error bars represent approximately the 5th and 95th percentile. Data was taken from a previous study (Messina et al., 2015). **b**, Yield difference between DT trait and non-DT trait vs. mean yield of non-DT trait based on our observed data.

Accordingly, we collected data from the western U.S. corn belt (**Fig. R2.2**) (Colorado, Kansas, and Nebraska; 8,992 data points across 36 field sites from 2000 to 2020). We excluded data in Texas because of limited accessible public datasets. We then plotted the yield difference between the mean yield of DT traits and ones of non-DT traits at the same first-trial year against mean non-

DT yields for each site-year (**Fig. R2.1b**). The results showed a similar pattern to previous findings (**Fig. R2.1a**), indicating that DT traits yield greater gains under lower-yielding conditions.

Fig. R2.2. Map of the study region across the western U.S. corn belt with circles representing field sites (Kansas, KS; Colorado, CO; and Nebraska, NE).

This pattern implies a positive relationship between yield gains and VPD, but it does not indicate a positive response of actual maize yield to high VPD. Under water deficit conditions, both LT and non-LT traits experienced yield reduction (**Fig. R2.3a**, derived from Gaffney et al., 2015). However, the LT trait mitigated these losses to some extent, resulting in higher yield in LT traits relative to non-LT traits (**Fig. R2.3a**). This was also reflected in our separate analysis of DT and non-DT traits, where both exhibited a negative response to high VPD during the grain filling period (**Fig. R2.3b**).

Fig. R2.3. a, Yield performance of AQUAmax (Dupont Pioneer) hybrids compared to non-AQUAmax commercial hybrids in on-farm strip trials across multiple water-limited and favorable environments. Error bars represent one standard deviation. Data was derived from the previous study (Gaffney et al., 2015). **b,** Yield response to the VPD level during the grain filling period for DT hybrids and non-DT hybrids based on our data. Error bars show 95% confidence interval.

To further quantify the relationship between actual yield and VPD, we combined the western data (8,992 data points) with our original data (92,096 data points) and re-ran the model. We found a negative response of yield to high VPD (**Fig. R2.4**), consistent with our findings in the Main text.

Fig. R2.4. Nonlinear effect of the vapor pressure deficit (VPD) on maize yield during the growing season based on data from the western and central U.S. corn belt. **a**, Effect of VPD on yield during the vegetative period (from planting to silking). Slopes to the left and right of VPD threshold (vertical dashed line) indicate the sensitivities to VPD below and above the threshold. The solid lines correspond to the ensemble average with colored shadow areas displaying the 2.5th – 97.5th percentile range of 1,000 bootstraps. **b**, Same as **a** but for the grain filling period (from silking to maturity).

Given that data from the western region constitutes less than 10% of total data points, the overall relationship between maize yield and high VPD may predominately reflect a relationship in the central U.S. corn belt. Therefore, we also conducted a separate analysis using only the western region data. Since daily VPD values are generally higher in the western region compared to the central region, we re-estimated the optimal VPD threshold for the western region. We found that the relationship between maize yield and high VPD remains negative (**Fig. R2.5**).

Fig. R2.5. Nonlinear effect of vapor pressure deficit (VPD) on maize yield during the growing season for the western U.S. corn belt. **a**, Effect of VPD on yield during the vegetative period (from planting to silking). Slopes to the left and right of the VPD threshold (vertical dashed line) indicate the sensitivities to VPD below and above the threshold. The solid lines correspond to the ensemble average with colored shadow areas displaying the 2.5th – 97.5th percentile range of 1,000 bootstraps. **b**, Same as **a** but for the grain filling period (from silking to maturity).

We appreciate your suggestion. These analyses verify your opinion that LT traits bring greater yield gains in western regions and also reinforce the robustness of our findings, highlighting the negative response of maize yield to high VPD. Given the consistent negative response between the combined data and our initial findings, we would like to retain the original results in the Main text.

We have concerns from a practical perspective whether integrating data from two distinct regions for combined analysis would be appropriate. For instance, agricultural management practices such as planting densities differ markedly between the two regions (**Fig. R2.6**). The western region has consistently maintained lower and relatively stable planting densities over the past two decades, whereas the central region has experienced a substantial increase (**Fig. R2.6**). This difference underlines the potential limitations of combining datasets from these regions, as doing so may mask actual relationships and reduce explanatory power.

Fig. R2.6. Time series of maize planting densities from 2000 to 2020 in the western (red) and central (blue) U.S. corn belt. Each dot is the year-specific average planting densities. The error bars represent one standard deviation.

Based on your suggestion, we added **Fig. R2.4** above and an additional figure (**Fig. R2.7 below**) in the modified Supplementary Information for supporting our main findings. These figures, developed from the combined dataset of the western and central U.S. corn belt, provided additional evidence in our study: (1) the negative response of maize yield to high VPD, and (2) the improvement in drought resilience of maize yield over the breeding advancement.

Fig. R2.7. Change in sensitivity of maize yield to the vapor pressure deficit (VPD) during growing season. **a**, Time series in sensitivity of yield to VPD over breeding during the vegetative period (VEG; from sowing to silking) and grain filling period (GFP; from silking to maturity). The solid lines represent the ensemble average with colored shadow areas displaying the 2.5th – 97.5th percentile range of 1,000 bootstraps. **b**, The sensitivity of yield to VPD for old hybrids (2000-2006), intermediate hybrids (2007-2013), and new hybrids (2014-2020). Bars represent the ensemble average with error bars showing the 2.5th – 97.5th percentile range.

Additionally, we added several sentences for these two figures as follows:

Lines 135-139: “This nonlinear response was also robust (Supplementary Fig. 10), even after including data from the western U.S. corn belt (Supplementary Fig. 11), a region typically characterized by drought-prone environments. The result highlights the detrimental impact of high VPD on maize yields, consistent with a previous study⁴⁰.”

Lines 173-175: “Furthermore, by expanding our analysis to include data from the western U.S. corn belt, we found a consistent improvement in drought resilience of maize yield over breeding progress (Supplementary Fig. 15).”

References:

- Cooper, M., Gho, C., Leafgren, R., Tang, T., & Messina, C. (2014). Breeding drought-tolerant maize hybrids for the US corn-belt: discovery to product. *J. Exp. Bot.* 65(21), 6191-6204.
- Messina, C. D., Sinclair, T. R., Hammer, G. L., Curan, D., Thompson, J., Oler, Z., ... & Cooper, M. (2015). Limited-transpiration trait may increase maize drought tolerance in the US Corn Belt. *J. Agron.* 107(6), 1978-1986.
- Gaffney, J., Schussler, J., Löffler, C., Cai, W., Paszkiewicz, S., Messina, C., ... & Cooper, M. (2015). Industry-scale evaluation of maize hybrids selected for increased yield in drought-stress conditions of the US Corn Belt. *Crop Sci.* 55(4), 1608-1618.

Developing a model that has broad applicability is necessary, yet not sufficient, to make predictions about impacts of climate change. As I indicate above, what the authors found as a negative yield response to water deficit in the central corn belt may turn out to be a positive relation when water deficits occur during grain filling, as one may expect in warmer and drier climates. I don't think the speculation about impacts of climate change is appropriate.

Because crop improvement is dynamic it is not sufficient to assess the impacts of climate change using a static model. While transpiration (and yield) response to vpd underpin drought tolerance in contemporary germplasm, other mechanisms may underpin drought tolerance in future cohorts of hybrids. Breeders select in a mixture of environments and test in a set of environments that is correlated to some degree. The degree and sign of the correlation dictates if how quickly the breeding programs can adapt to the changing climate. A recent study shows how Corteva breeding adapted the germplasm despite the changing climate (<https://www.biorxiv.org/content/10.1101/2023.09.19.558447v1>). During this period many physiological traits changed underpinning the adaptation of maize to drier conditions in the western corn belt and wetter conditions in the central corn belt. This section of the manuscript needs to be revised.

Response: Thank you for your thoughtful comment and for sharing the recent work from Messina et al. (2023). We agree with your thoughts about the impact of climate change. Developing models with broad applicability is indeed important for estimating the impacts of climate change. We have integrated data of the western and central U.S. corn belt for additional analysis. As demonstrated in our previous response, the negative response of maize yield to high VPD remains consistent, reinforcing the robustness of our findings.

We agree that static models can introduce uncertainties when projecting future climate impacts, as they assume that historical relationship between variables remains unchanged - a simplification that may not capture the evolving physiological and genetic traits driven by breeding programs. As highlighted in the work by Messina et al. (2023), the dynamic nature of breeding programs is evident in the significant improvement of genetic gains from 1990 to 2020, reflecting an adaptatively dynamic response to changing climate.

Accounting for these dynamic responses represents a major challenge, as the future interaction between crop traits and climate remains uncertain. We recognize that integrating the evolving nature of crop improvements into future climate impact estimates is an important next step. Thus, we have revised the Main manuscript to acknowledge this limitation and emphasize the need for advanced approaches that can better address this complexity in future studies, as follows:

Lines 278-284: “A limitation of our projections of future climate change impacts is that they are based on static historical relationships between maize yield and VPD. While this approach provides valuable insights, highlighting continuous threats of atmospheric drought in the future may oversimplify the complex, evolving interactions likely to occur under future conditions, particularly given the inherently dynamic nature of the interaction between crop trait improvement and environment change⁵³. Future studies should prioritize developing models that incorporate evolving crop traits and breeding strategies to more accurately assess future climate impacts.”

References:

Messina, C. D., Borrás, L., Tang, T., & Cooper, M. (2023). Crop improvement can accelerate agriculture adaptation to societal demands and climate change. *BioRxiv*, 2023-09.

A minor comment relates to the effect of CO₂ fertilization in maize. Due to the C₄ photosynthesis pathway, it is unlikely that maize yields will respond to CO₂ directly but through an improved water use efficiency.

Response: This is an insightful point, and we agree. We have revised the Discussion section as follows:

Lines 284-289: “Additionally, our analysis does not account for the potential effect of CO₂ fertilization on maize yields. Although maize is less responsive to elevated CO₂ concentrations in terms of direct yield gains due to the C₄ photosynthesis pathway, elevated CO₂ can enhance maize performance indirectly by reducing stomatal conductance, thereby improving water use efficiency, particularly under drought-prone conditions⁵⁴.”

Reference:

Morison, J. I. (1985). Sensitivity of stomata and water use efficiency to high CO₂. *Plant, Cell & Environ.* 8(6), 467-474.

-End of Response for Referee 2.

Reviewer #3 (Remarks to the Author):

Zhao et al. described a comprehensive study to analyze the yield trials of U.S. Corn Belt and environmental data to draw a conclusion about the genetic improvement of corn at a large scale. Authors carefully collected available data from 3 key corn production states and conducted systematic analyses. At the first read, it did not appear to be anything special other than a large data set. However, upon reflection, I realized that this was indeed a great way to identify patterns from the data and there was no other feasible way to run new experiments to test this. Any designed experiments would not be able to capture the actual hybrid-environment combinations happened in the past at such an extensive trial scale, including that earlier era hybrid study from one major company, the ongoing era hybrid study from a different company, or the ongoing G2F initiative. The analysis procedure makes sense and the conclusion is very justified. This study has significant impact on setting the record straight than a previous small-scale study (marginal corn production area), and correctly reflect what has been going on in crop improvement and production. I commend authors for working on this important research area.

Response: Thank you for your encouraging comments and insights. Your comments are valuable in helping us improve the clarity and quality of the manuscript. Our point-by-point response to all your comments are below.

I have the following comments for authors consideration.

1) It would be helpful to have some generic diagrams as supplementary figures to support the main figures and to show the several important concepts in the study. Text description does not always do a good job. “the release year-specific average yield”, how “vegetative” and “grain-filling period” were identified for hybrid/site-year, how the compositions of three groups of hybrids change along the year (2000-2020) since they only appeared in some years, how often DT hybrids landed in one of the three groups, and the existence of LYH-MYH-HYH, the old-median-new, and nonDT-DT asks for any further sub-setting if possible (?).

Response: Thank you and we agree. We provided a response below and added these associated figures as the Supplementary Information (SI) for each point (P) you suggested:

P1. “the release year-specific average yield”

The term “the release year-specific average yield” has been replaced with “first-trial year average yield” in our revised manuscript for accuracy and consistency. The “first-trial year” refers to year in which the hybrids first appeared in the trials, as noted in Line 368 in Main text.

We then added a diagram (**Fig. R3.1**) to our modified SI to clarify the calculation process in the first trial-year average yield. Firstly, for each hybrid, we identified the initial year it appeared in the field trials. We then extracted the corresponding yield data from the first trial year. Finally, we calculated the average yield across all hybrids introduced in the same first trial year, which we refer to as the “first trial-year average yield.”

Fig. R3.1. Workflow to calculate the first trial-year average yield.

P2. “how “vegetative” and “grain-filling period” were identified for hybrid/site-year.”

The vegetative period is defined from planting (PT) to silking (SK), while the grain-filling period spans from SK to maturity (MT) for each site-year. PT and harvest (HV) dates are available for each site-year, while SK and MT were not directly recorded. To estimate these missing phenological stages (SK and MT), we integrated site-year phenological data with state-level maize phenological progress reports from the United States Department of Agriculture (USDA-NASS), which provided annual PT, SK, MT, and HV dates for each state.

To clarify the process to estimate SK and MT, we took the Belleville site in Illinois (IL) as an example (**Fig. R3.2**). Firstly, based on state-level phenological data in IL from USDA-NASS (**Fig. R3.2a**), we calculated growing degree days (GDD; °C days) for three phenological periods: PT-SK; PT-MT; and PT-HV (**Fig. R3.2b**). The GDD was calculated through an adjusted daily air temperature [$GDD = \sum_i (T_i^{adj} - 10)$]. T_i^{adj} is daily adjusted temperature (°C) on day i , which was calculated by (1) imposing a maximum of 30°C and a minimum of 10°C on the daily minimum and maximum temperature, and (2) taking the average of these adjusted minimum and maximum temperatures (Sacks and Kucharik, 2011). The “10” in the equation refers to base temperature for maize to growth. Second, we calculated the state-level ratio of GDD for PT-SK and PT-MT relative to total GDD from PT to HV (GDD_{PT-HV}) (**Fig. R3.2c**). Thirdly, we used specific site-year PT and HV from field trials (**Fig. R3.2d**) to calculate GDD_{PT-HV} for Belleville (**Fig. R3.2e**). Fourth, assuming that the GDD ratio for a specific phenological period is consistent across sites within a state, we then estimated GDD_{PT-SK} and GDD_{PT-MT} for the Belleville site (**Fig. R3.2f**) by multiplying the state-level GDD ratio (**Fig. R3.2c**) with Belleville’s GDD_{PT-HV} (**Fig. R3.2e**). Finally, we estimated the silking and maturity dates based on the estimated GDD requirements (**Fig. R3.2g**).

To evaluate the accuracy of our approach, we compared the estimated with available observed silking dates from field sites. The estimates showed a bias of 5 days (root mean square error; RMSE) (**Fig. R3.3**), indicating the reliability of our estimation method.

Based on your suggestion, we added **Fig. R3.2** and **Fig. R3.3** in the modified SI.

Fig. R3.2. Flow of estimating silking and maturity dates from 2000 to 2020 using the Belleville site in Illinois (IL) as an example. **a**, Observed state-level phenological periods derived from the USDA dataset. **b**, State-level growing degree days (GDD) required for specific phenological periods, including from planting (PT) to silking (SK) (GDD_{PT-SK}), from PT to maturity (MT) (GDD_{PT-MT}), and from PT to harvest (HV) (GDD_{PT-HV}). **c**, The ratio of GDD for specific phenological periods relative to GDD_{PT-HV} . **d**, Observed PT and HV. **e**, GDD required from PT and HV. **f**, Estimated GDD_{PT-SK} and GDD_{PT-MT} based on panel c and panel e. **g**, Estimated annual silking and maturity dates.

Fig. R3.3. Estimated vs. observed silking date of maize from field sites in Ohio from 2000 to 2012. The solid line refers to the fitted line. RMSE refers to root mean square error, showing the estimated performance. N represents the number of samples.

Reference:

Sacks, W. J., & Kucharik, C. J. (2011). Crop management and phenology trends in the US Corn Belt: Impacts on yields, evapotranspiration and energy balance. *Agric. For. Meteorol.* 151(7), 882-894.

P3. “how the compositions of three groups of hybrids change along the year (2000-2020) since they only appeared in some years.”

We categorized maize hybrids into three groups (low-, median-, and high-yielding hybrids) for each site-year based on local percentile thresholds of maize yields. This approach ensures that hybrids of all three groups at a given site-year experience the same agricultural management practices and environments. Consequently, all three hybrids groups for each site-year appeared in each single year from 2000 to 2020.

To clarify our classification approach, we added **Fig. R3.4** below in the modified SI using the year 2000 at the Belleville site as an example. In this case, we had yield data for 63 hybrids. We first calculated the 25th and 75th percentiles of hybrid yield for this site-year. Next, we divided hybrids into three groups: low-yielding (below the 25th percentile), median-yielding (between the 25th and 75th percentiles), and high-yielding (above the 75th percentile).

We repeated this process for each site-year. Therefore, we had data from three groups from 2000 to 2020.

We also added a sentence into the method section, as follows:

Lines 310-312: “A case using hybrid yield data in 2000 at Belleville, Illinois for dividing maize hybrids into three yielding types is shown in Supplementary Fig. 22).”

Fig. R3.4. A case (year 2000 for the Belleville site) for dividing maize hybrids into three yielding types, including low-yielding, median-yielding, and high-yielding hybrids. The yield performance of 63 maize hybrids was measured in 2000 at the Belleville site. The x-axis refers to yield, and y-axis represents hybrids. Black points are actual maize yield for each hybrid. Vertical dash lines represent the 25th and 75th percentiles for defining the boundary of three yielding classes.

P4. “how often DT hybrids landed in one of the three groups.”

Thank you for this great question. Regarding DT hybrids, we only evaluated whether DT hybrids showed greater drought resilience compared to non-DT hybrids as provided in the Supplementary Information (Fig. S16). Due to relatively less data in DT hybrids, we did not focus on it much in this study.

In terms of your point, the time series in fraction of DT hybrids in the total dataset for three groups (**Fig. R3.5**) are shown. DT hybrids were gradually introduced after 2011 and evenly distributed among three groups.

Fig. R3.5. Fractions of DT hybrids among all tested hybrids in the specific first-trial year, categorized by yield type.

P5. “the existence of LYH-MYH-HYH, the old-median-new, and nonDT-DT asks for any further sub-setting if possible (?)”

Regarding further sub-setting, our analysis also included some interacted groups among these categories. For example, we assessed whether maize hybrid advancement (from old to new) consistently improved drought resilience among three yielding categories (LYH, MYH, and HYH), shown in Fig. 4a in the Main text and Fig. S9 in the modified SI.

Additionally, based on your comment 6 below “6. Fig. 4a, would it be good to use the natural production average to partition the data into two groups, those “normal years” to obtain the slope and those above and below the trend line years. But I suspect it would cancel out? Even this is the case, feels like establishing some sort of connection with the production would be beneficial to this study,” we also included a category for bad-, normal-, and good-production conditions in the modified manuscript.

2. Fig. S5 did not seem to provide clear and adequate justification for the selection of these two thresholds in Fig. 2. I was expecting more evidence as this part is very critical to the analysis of this comprehensive dataset.

Response: Thank you for your thoughtful comment. To determine the vapor pressure deficit (VPD) thresholds selected in Fig. 2, we employed a piecewise linear regression approach, following Schlenker and Roberts (2009). This approach allows for one threshold per phenological stage, thus detecting potential nonlinear yield response to VPD. We assessed a range of candidate thresholds and estimated the performance of the corresponding models. The final thresholds were selected based on the lowest Akaike Information Criterion (AIC) value. AIC is a widely accepted metric for model comparison, where lower values indicate a better fit (Akaike, 1998). When the difference

in AIC values between two models is less than 2, it suggests that the models perform similarly well (Burnham et al., 1998).

The candidate threshold window was determined using the distribution of daily VPD during each phenological stage. We restricted the threshold range from the 50th to 90th percentile range (**Fig. R3.6a,b**) to ensure coverage from median to extreme stress levels while minimizing noise from the tails of the distribution. This corresponds to a range of 1.3-2.1 kPa for the vegetative (VEG) stage and 1.3-1.9 kPa for the grain filling stage (GFP). To improve computation efficiency, we discretized this range into 49 threshold pairs, ranging from 1.3 kPa to 1.9 kPa in increments of 0.1 kPa (**Fig. R3.6c**). For each pair, we constructed two measures of VPD exposures (below and above threshold) for each phenological stage and re-estimated the performance of the full model. The optimal VPD thresholds (the lowest AIC) were estimated to be 1.4 kPa for VEG and 1.3 kPa for GFP (**Fig. R3.6d**). This approach was also worked well in previous studies to estimate temperature threshold impacting the U.S. wheat yield (Tack et al., 2015).

To evaluate the robustness of our threshold selection, we extended the bottom bound of the threshold range to 1.1 kPa, equivalent to the 30th percentile of daily VPD distribution, and repeated the analysis using 81 threshold pairs (**Fig. R3.7**). While the threshold combination with the lowest AIC slightly shifted (1.4 kPa for VEG; 1.2 kPa for GFP), our original thresholds remained the second-best fit, with $\Delta\text{AIC} < 2$ (**Fig. R3.7**). This indicates no statistical difference in model performance, supporting our initial selection.

Given that our two main findings in this study include: 1) the nonlinear response of maize yield to VPD during both phenological stages (Fig. 2 in Main text); and 2) the improved drought resilience of hybrids during the grain filling period (Fig. 4a in Main text), we further tested the robustness using three additional near-optimal thresholds combinations (blue cross; **Fig. R3.6d**). Results remained consistent (**Fig. R3.8**), reinforcing the confidence of our conclusions.

Based on your suggestion, we replaced the original Fig. S5 with **Fig. R3.6** and added text and **Fig. R3.8** to our revised SI to clarify the selection procedure and highlight the rationale behind the chosen thresholds. Additionally, we also added two sentences to the results section regarding the robustness of our findings when using different thresholds, as follows:

Lines 133-135: “To test the robustness of our findings, we used three additional near-optimal thresholds combinations (blue cross; Supplementary Fig. 7). The nonlinear effect of maize yield to VPD remained consistent (Supplementary Fig. 9).”

Lines 172-173: “This remained robust when using three additional near-optimal thresholds combinations (Supplementary Fig. 9c,f,i).”

References:

- Schlenker, W., & Roberts, M. J. (2009). Nonlinear temperature effects indicate severe damages to US crop yields under climate change. *Proc. Natl. Acad. Sci. U.S.A.* 106(37), 15594-15598.
- Akaike, H. (1998). Information theory and an extension of the maximum likelihood principle. In *Selected papers of hirotugu akaike* (pp. 199-213). New York, NY: Springer New York.
- Burnham, K. P., Anderson, D. R., Burnham, K. P., & Anderson, D. R. (1998). *Practical use of the information-theoretic approach* (pp. 75-117). Springer New York.

Tack, J., Barkley, A., & Nalley, L. L. (2015). Effect of warming temperatures on US wheat yields. *Proc. Natl. Acad. Sci. U.S.A.* 112(22), 6931-6936.

Fig. R3.6. Flow chart for the procedure of selecting the vapor pressure deficit (VPD) threshold. **a**, The distribution of daily VPD during vegetative stage (VEG). Vertical dashed lines represent the selection window for the VPD threshold, spanning from the 50th to the 90th percentile. **b**, Same as panel **a**, but for the grain filling period (GFP). **c**, Matrix of 49 paired VPD threshold combinations for [VEG; GFP]. **d**, The Akaike Information Criterion (AIC) values for each threshold group G1 to G49.

Fig. R3.7. Change in Akaike Information Criterion (AIC) across vapor pressure deficit (VPD) threshold groups G1 to G81. Each threshold group shown in the right box represents a specific combination of VPD thresholds [vegetative stage (VEG), grain filling stage (GFP)].

Fig. R3.8. The nonlinear effect of vapor pressure deficit (VPD) on maize yield and the estimated change in drought resilience of hybrid advancement during the vegetative (VEG) (green) and grain filling period (GFP) (orange) using three combinations of VPD thresholds, including [1.3 kPa (VEG), 1.4 kPa (GFP)] (a-c); [1.5 kPa (VEG), 1.3 kPa (GFP)] (d-f); and [1.6 kPa (VEG), 1.3 kPa (GFP)] (g-i).

3. Fig. 3a-c and associated analyses and results need more attention. You did two way grouping based on the marginal means, site-year mean and hybrid mean, and generated the plots. Not sure why the test was conducted. And if $p > 0.05$, does not mean there is no enough evidence for your separation.

Response: Thank you for your comment. The goal of this analysis was to assess whether breeding progress-driven yield trends exist statistically significant difference across varying environmental conditions. For example, if yield gains at the worst yield condition is not significantly different from those in optimal conditions, this would suggest that breeding progress contributes to improved resilience to adverse environment conditions, enabling relatively ‘parallel gains in maize yield and drought resistance’ across a range of environmental conditions. Therefore, we assessed the statistical difference of yield trends among varied environmental conditions. We found $p > 0.05$, indicating a parallel yield gain over breeding across diverse environmental conditions.

We agree that $p > 0.05$ should not be interpreted as proof of absence of separation. The error bars (Fig. 4a-c) and consistent trend directions (Fig. 4d-f) provide additional context. To clarify the objective of conducting this test, we modified one sentence in the result section and added several sentences to the method section, as follows:

Lines 154-155: “the difference was not statistically significant ($p > 0.05$), indicating parallel yield gains over breeding across a range of environments.”

Lines 349-353: “To assess whether yield trend over breeding show statistically significant difference across varying environmental conditions, we performed a two-sided Student’s t test at a 95% confidence level. A p-value greater than 0.05 would suggest that breeding progress contributes to relatively parallel gains in maize yield across a range of environmental conditions.”

Also, check you statistics, “Bars refer to the mean trend, and error bars are 95% confidence interval”. The standard error for the mean is generally very small with large sample size, and the 95% confidence interval for the mean would not be that wide.

Response: Thank you for your insight. You are correct that the standard error for the mean is generally small for a large sample size. However, in our analysis, the error bars represent the 95% confidence interval (CI) of the estimated yield trend, calculated as $1.96 \times$ standard error (SE) of the regression slope, derived from a linear fit. Our data included 21 data points (one value for each year from 2000 to 2020), which is a relatively small sample size, and we have confirmed that the CI width is reliable.

To clarify our approach, we have added a figure (Fig. R3.9 below) to our modified SI, using low-yielding hybrids (LYH) as an example. We first calculated the average yield of hybrids for each site-year (Fig. R3.9a), representing site-year environment index. For each year, we categorized all sites into five environmental stress levels based on yield percentiles (10th, 25th, 75th, and 90th percentiles) (Fig. R3.9b-c), and calculated the average yield across sites for each environmental stress level (Fig. R3.9d). Next, we fitted a linear regression of yield vs. years for each stress level (Fig. R3.9d) and extracted the slope and its 95% CI (Fig. R3.9e). Lastly, we plotted the estimated

trends and their CIs for all environmental stress levels (**Fig. R3.9f**), corresponding to Fig. 3a in the Main text.

Overall, the error bars in our plot reflected the uncertainty in the estimated trend slopes over time, capturing the variability in yield trends across environmental stress levels. Based on your insights, we added **Fig. R3.9** to our modified SI to clarify our approach for the trend analysis.

Fig. R3.9. Yield trends across varying environmental stress level. **a**, Site-level mean yield for each year. Points represent sites. **b**, Yield percentiles across all sites for each year. The lower percentile represents the worse environmental level. **c**, Same as panel **a**, but points were classified into specific environmental levels. **d**, Time series of mean yield over years. **e**, Estimated yield trends and their 95% confidence intervals (CIs). **f**, Yield trends and their 95% CI for all environmental stress levels.

4. Fig. 3d-f, please indicate the difference between the red (base lines with the fixed slope 1.0s, each group of hybrids by themselves (check?)) and the rest. What was plotted and the legend text, “Dashed lines show the fitted linear regression for each environmental index level” are not easy to understand, since you introduced the year here. You probably do not have to do a comprehensive analysis to declare the significant difference among those slopes. But I hope some of the slopes are different from others.

Response: Thank you. In an absolute sense, we found that breeding progress over the past two decades has improved yield at a relatively consistent rate across diverse environmental stress levels (Fig. 3a-c in the Main text). To further explore this point, we analyzed relative yields (Fig. 3d-f in the Main text), calculated as the ratio of the actual yield at specific environmental levels (Fig. R3.10a below) to the trend yields under the best conditions which serves as the reference baseline (red fitted line in Fig. R3.10a).

The solid and dashed lines in Fig. 3d-f (or Fig. R3.10b) represent the changes in fractional yield and their fitted trends, respectively, across diverse environmental stress levels. If breeding progress had disproportionately favored higher-yielding environments, we would expect declining trends (negative slopes) in fractional yields under stressed conditions. However, we observed non-negative slopes, and these values of slopes closed to 1 across all environmental levels, indicating that breeding progress has parallelly contributed to yield improvements across all environmental conditions.

Based on your question, we added several sentences to the method section and incorporated Fig. R3.10 into the modified SI to clarify our approach for Fig. 3d-f in the Main text, as follows:

Lines 354-358: “To further explore this point, we analyzed relative yields, calculated as the ratio of the actual yields at specific environmental levels to the trend yields under the most favorable conditions, which serves as a reference baseline (Supplementary Fig. 24). If breeding progress had disproportionately favored higher-yielding environments, we would expect declining trends (negative slopes) in fractional yields under stressed conditions.”

Additionally, in the revised manuscript, we included a legend for Fig. 3d-f and rewrote the figure caption to improve clarity.

Fig. R3.10. Time series of maize yield across different environmental conditions from the worst (blue line) to the best environmental condition (red line). **a**, Time series of actual mean yield. The straight lines refer to trend yield, derived from the least square regression. The red trend line refers to the trend under the best environment conditions, which is defined as the reference baseline. **b**, Same as panel **a** but expressed as fractional yields, calculated as the ratio of actual yield to the trend yield under the best environment conditions. Dashed lines represent the fitted fractional yield, derived from the least square regression.

Fig. R3.11 (our revised caption for Fig. 3 in Main text). Trend of maize yield across five environmental levels from 2000 to 2020 for low-yielding hybrids (LYH), median-yielding hybrids (MYH), and high-yielding hybrids (HYH). **a-c**, Trends of maize yield across five environmental levels from the highest environmental index level (red color), representing the best yield conditions, to the lowest environmental index level (blue color), indicating the worst yield conditions. Bars refer to the mean yield trend, and error bars are the 95% confidence interval. **d-f**, Time series of fractional yields (relative to the best environmental conditions) for LYH, MYH, and HYH, respectively (see Methods). The solid lines show mean fraction yield, and dashed lines indicate fitted linear trends for five environmental index levels.

5. Fig. 4a, any reasons to explain the wild swings of the sensitivity? I think authors know the documented rainfall and temperature patterns that led to this.

Response: Thank you. This is a thoughtful observation. Understanding the fluctuations in yield sensitivity to high VPD (VPD above the threshold) (Fig. 4a) is important for identifying mechanisms behind vulnerability.

To investigate potential climate-related swings, we conducted linear regressions between detrended VPD sensitivities and detrended sub-seasonal climate variables, including cumulative precipitation and mean maximum temperature during the vegetative and grain filling periods (Fig.

R3.12). These analyses did not yield statistically significant relationships ($p\text{-value} \geq 0.10$), suggesting that average sub-seasonal temperature and precipitation alone may not explain the year-to-year variability in VPD sensitivity (because our modeling Eq. 7 in the Main text includes temperature and precipitation predictors).

Nonetheless, specific years such as 2011, 2012, 2013, and 2020 correspond to dry conditions during grain-filling periods across the U.S. Corn Belt and align with sharp declines in VPD sensitivity, particularly for GFP in Fig. 4a. suggesting a dip sensitivity for GFP (Fig. R3.12 and Fig. 4a). Both maximum temperature (as we weighted the VPD estimate for daytime and heat stress), and lower sub-seasonal precipitation (so that low soil moisture) often align with yield dips as well as accompanying more negative VPD sensitivities. These years were also characterized by sub-seasonal precipitation deficits and extreme daytime temperatures, which likely contributed to soil moisture depletion and increased heat stress. These combined conditions are consistent with reduced stomatal conductance and carbon assimilation, magnifying the negative impacts of high VPD on yield.

Fig. R3.12. The relationship between the high VPD sensitivity and climate variables during the vegetative (VEG) and grain filling period (GFP), including cumulative precipitation and mean maximum temperature. Solid lines refer to the fitted line based on the least square regressions.

Any need to calculate the slopes for the shaded areas?

Response: Thank you for your question. The shaded area in Fig. 4a spans from 2012 to 2020, covering a relatively short time window (less than a decade). While there was a noticeable upward trend in sensitivity during this period, with an estimated slope of $0.32 \text{ t ha}^{-1} \text{ kPa}^{-1} \text{ yr}^{-1}$ (95% confidence interval: $0.25\text{--}0.39 \text{ t ha}^{-1} \text{ kPa}^{-1} \text{ yr}^{-1}$), we chose not to emphasize this value. Estimating trends over such a short time frame may overstate the long-term trajectory of hybrid advancements in drought resistance. We appreciate your observation.

Not sure whether some un-adapted hybrids from some companies outside of U.S. were removed from the analysis.

Response: Thank you for raising this point. Prior to your comment, we had not explicitly focused on the geographic origin of the hybrids included in our analysis, as the primary goal of the performance trials was to evaluate hybrids under a wide range of U.S. environmental conditions to assist farmers in selecting suitable hybrids for their regions. As such, we initially assumed that the tested hybrids were sourced from U.S. companies.

We checked the original “hybrids performance test” reports, which document the source of each hybrid. We confirmed that all hybrids included in our dataset were provided by U.S. companies. For example, over the past two decades, all hybrids tested in Iowa originated from 92 companies located in the U.S. (**Table R3.1**). Therefore, our dataset did not include hybrids from non-U.S. sources.

Based on your question, we added one sentence into the Method section, as follows:

Line 299: “The sources of all hybrids were provided by U.S. companies.”

Table R3.1. The maize hybrids tested in Iowa were provided by 92 companies located in the U.S. including Iowa (IA), Illinois (IL), Indiana (IN), Michigan (MI), Minnesota (MN), Missouri (MO), Nebraska (NE), North Carolina (NC), and Wisconsin (WI).

Sources of maize hybrids		
Adler Seeds, Sharpsville, IN	GES Inc., Boone, IA	Mycogen Seeds, Indianapolis, IN
Adrian Associates, Galesburg, MI	Genetic Enterprises Int'l., Johnston, IA	NK Brand: Syngenta, Minnetonka, MN
Ag Com, Inc., Essex, IA	Gold Country Seed, Hutchinson, MN	NorthStar Genetics, Albert Lea, MN
AgReliant Genetics, Elmwood, IL	Golden Harvest Seeds, Pekin, IL	NuTech Seed, Forest City, IA
AgSource Seeds, Ames, IA	Golden Harvest: Syngenta, Minnetonka, MN	NuTech Seed, LLC, Ames, IA
AgSource Seeds, Nevada, IA	Great Lakes Hybrids, Ovid, MI	NuTech Seeds, Ames, IA
Albert Lea Seed, Albert Lea, MN	GrowDirect, Inc., Monticello, IN	Pfister Hybrids, El Paso, IL
Banks Seeds, Boone, IA	Hawkeye Hybrids, Inc., Pella, IA	Phoenix: Beck's Hybrids, Atlanta, IN
Beck's Hybrids, Atlanta, IN	Hi Fidelity Genetics, Durham, NC	Pioneer Hi-Bred, Int'l., Johnston, IA
Blue River: Blue River Hybrids, Ames, IA	Hobart Brothers Seeds, Lake City, IA	Pioneer: Corteva Agriscience, Johnston, IA
Cappel Certified Seeds Inc., Rochelle, IL	Hoegemeyer Hybrids, Hooper, NE	Pioneer: DuPont Pioneer, Johnston, IA
Champion Seed, Ellsworth, IA	Integrity Hybrids, Kelley, IA	Prairie Hybrids, Deer Grove, IL
Channel Bio Corp, Kentland, IN	Iowa State University, Ames, IA	Premium Seed, Inc., Berwick, IL
Channel, Huxley, IA	Jacobsen Hybrid Corn Co., Inc, Lake View, IA	Producers Hybrids, Battle Creek, NE
Circle Seed Co., Dike, IA	Jung Seed Genetics, Randolph, WI	Rainbow Seeds, Inc., Oskaloosa, IA
Cornelius Seed, Bellevue, IA	Kaltenberg Seeds, Waunakee, WI	Renk Seed Co., Sun Prairie, WI
Crow's Hybrid Corn Co., Kentland, IN	Kruger Seed Co., Dike, IA	Renze Hybrids, Inc., Carroll, IA
DEKALB: Monsanto, St. Louis, MO	LG Seeds, Elmwood, IL	Roeschley: Miller Hybrids, Inc., Kalona, IA
Dairyland Seed Co., Clinton, WI	LG Seeds, Jefferson, IA	Spectrum Seed Solutions, Darlington, IN
Merschman Seeds, Inc., West Point, IA	LG Seeds, Westfield, IN	Spectrum Seed Solutions, Linden, IN
Dow AgroSciences, Indianapolis, IN	Lewis Hybrids, Inc., Ursa, IL	Syngenta Seeds, Ames, IA
DuPont Pioneer, Johnston, IA	Longping High-Tech Seeds, LLC, Capron, IL	Titan Pro SCI, Inc., Clear Lake, IA
DuraCrop: DuraCrop Seed, Oskaloosa, IA	MFA Inc., Columbia, MO	Trelay Seed Co., Livingston, WI
DuraCrop: Rainbow Seeds, Oskaloosa, IA	Masters Choice Inc., Anna, IL	Trisler Seeds, Inc., Fairmount, IL
Dyna-Gro Seed, Pocahontas, IA	Merschman Seeds, Inc., West Point, IA	Unity Ag Direct, West Lafayette, IN
Epley Bros. Hybrids, Inc., Shell Rock, IA	Middlekoop Seed Corn, Inc., Packwood, IA	Unity Seeds, Kentland, IN
Farm Advantage Corp., Belmond, IA	Midwest Seed Genetics, Inc., Carroll, IA	Unity Seeds, LLC, Lafayette, IN
Fontanelle Hybrids, Fontanelle, NE	Miller Hybrids, Inc., Iowa City, IA	Viking, Albert Lea, MN
Fontanelle Hybrids, Fremont, NE	Miller Hybrids, Inc., Kalona, IA	Willcross Hybrids, Garden City, MO
Four Star Seed Co., Logan, IA	Monsanto, St. Louis, MO	iCORN.com, Cicero, IN
G2 Genetics (NuTech), Ames, IA	MorCorn: MFA Inc., Columbia, MO	

Also, please correct this statement, “We found a slight decrease ($P > 0.10$) in maize yield resistance to drought during the VEG period,” You cannot have both ways. Not significant, so no change.

Response: Thank you for pointing this out. We modified the following sentence,

Lines 168-169: “We found no statistically significant change in maize yield resistance to drought

during the VEG period ($p > 0.10$).”

6. Fig. 4a, would it be good to use the natural production average to partition the data into two groups, those “normal years” to obtain the slope and those above and below the trend line years. But I suspect it would cancel out? Even this is the case, feels like establishing some sort of connection with the production would be beneficial to this study.

Response: We appreciate your insight. This is an excellent idea and provides a new perspective on how breeding advancements may have influenced drought resistance of maize yield under different production conditions.

Following your insights, we conducted a linear regression between trial year and average yield for each state (**Fig. R3.13a**). Based on fitted yield trend and its 95% confidence interval (CI), we then categorized years into three groups: good, normal, and bad years. Specifically, years where observed mean yield exceeded the upper CI were classified as good years (blue points; **Fig. R3.13a**), those below the lower CI as bad years (red points; **Fig. R3.13a**), and those within the CI range as normal years (gray points; **Fig. R3.13a**).

Next, we split the original dataset into three production conditions (**Fig. R3.13b**) and re-estimated the VPD sensitivity for each data subset using the same linear mixed-effects modeling framework as described in the Main text (**Fig. R3.13c**). During the vegetative stage (VEG), we found no statistically significant change for three production conditions ($P > 0.10$; left panels in **Fig. R3.13c**). However, during the grain filling period (GFP), a clear gradient emerged: VPD sensitivity did not significantly change in good years ($P > 0.10$), but showed a statistically significant improvement in bad years ($P = 0.01$), and a marginally significant increase in normal years ($P = 0.06$) (right panels in **Fig. R3.13c**). These findings suggest that breeding progress over the past two decades has improved drought resistance during the GFP period, especially under unfavorable production conditions.

Based on your suggestion, we added **Fig. R3.13** below to our modified SI and incorporated several sentences into our revised Main text.

Lines 187-197: “Additionally, we assessed how breeding advancements may have influenced drought resistance of maize yield under different production conditions. We categorized trial years into three production conditions for each state: good, normal, and bad years (see Supplementary Text 3 and Supplementary Fig. 18). We found no statistically significant change ($p > 0.10$) in drought resistance during the VEG stage (Supplementary Fig. 18), consistent with our findings (Fig. 4a). However, during the GFP period, a clear gradient emerged: VPD sensitivity did not significantly change in good years ($p > 0.10$) but showed a statistically significant improvement in bad years ($p = 0.01$), and a marginally significant increase in normal years ($p = 0.06$) (Supplementary Fig. 18). These findings suggest that breeding progress over the past two decades has improved drought resistance during the GFP period, especially under unfavorable production conditions.”

Fig. R3.13. The framework to estimate the change in high vapor pressure deficit (VPD) resistance of maize yield over breeding progress under different production conditions. **a**, State-level time series of maize yields. Solid lines refer to regressed yields (dark purple) with 95% confidence interval (light purple). Blue points refer to good years, gray points are normal years, and red points represent bad years. **b**, Flowchart for data classification and modeling. **c**, Time series in sensitivity of yield to VPD during the vegetative period (VEG; from sowing to silking) and the grain filling period (GFP, from silking to maturity).

7. Figs. 4c and the text. This VPD increase was for what stage, GFP?

Response: Thank you. The VPD increase was for the entire growing season, including the vegetative (VEG) and grain filling period (GFP). To clarify this, we revised the figure caption and modified one sentence in the method section as follows:

Lines 430-431: "...during the growing season, including the VEG and GFP periods."

8. Not criticism. The compiled data needs to be made available in the final stage. Indicating the source is not adequate for checking and re-analysis.

Response: We agree and thank you for your suggestion. We will provide original data in the final stage, following the journal's data availability policy.

-End of Response for Referee 3.

Point-by-point responses to three Referees

The authors have used regular fonts for the Referees' comments, blue fonts for our responses, and red fonts with quotation marks to show the revised text.

Referee 1

Reviewer #1 (Remarks to the Author):

The authors have provided a comprehensive response to the comments, suggestions and questions raised by the three reviewers. I commend the authors on providing a well-structured and informative response document that explains the improvements made to their manuscript.

Response: We appreciate for your encouraging comments and insights. Your comments are valuable in helping us further improve the quality of the manuscript.

One final aspect that the authors could include to further improve their manuscript is clarify the importance of long-term selection for integrated Hybrid-by-Management technology combinations for the long-term maize productivity gains and reduction in hybrid sensitivities to environmental conditions. This has been well documented by Duvick in his many studies of long-term genetic improvement of hybrids from commercial breeding targeted at the US corn-belt.

Response: In response to your insight, we reinforced the Main text accordingly, as follows:

Lines 273-276: “These findings are also consistent with previous studies^{18,52}, which emphasized the importance of long-term selection through integrated hybrid-by-management technology combinations in sustaining maize productivity gains and reducing hybrid sensitivities to environment conditions.”

While the authors' data set does not provide a structured sequence of experiments across years with a factorial of hybrids and agronomic management strategies (such as the experiments conducted by Duvick and colleagues), the authors have done some inciteful breakouts of good, normal and bad years (Fig. R3.13) in response to a suggestion by Reviewer 3. This breakout revealed evidence of an improvement (reduced sensitivity) to VPD in bad years for the GFP. This is an important result. The authors can connect this finding from their analyses with the results reported as Figure 8.21 in the cited publication by Cooper, M. et al. Predicting Genotype× Environment× Management (G× E× M) interactions for the design of crop improvement strategies: integrating breeder, agronomist, and farmer perspectives. *Plant Breeding Reviews* 46, 467-585 (2022). In Figure 8.21 within Cooper et al. (2022), and the associated text, these authors demonstrate the sequence of improvements in hybrid yield performance under drought conditions, achieved by shifting hybrid water use patterns by breeding, resulted in reduced the sensitivities of hybrids to droughts during flowering time and grain filling periods. This allowed farmers to shift their agronomic management to higher plant densities in combination with changes in irrigation management (for the Western regions) to further protect hybrids during the grain filling period. The trends in improvements in GFP performance of hybrids under stress that the authors have revealed, their Fig. R3.13 from re-analysis of their data sets, are consistent with the shifts in hybrid-by-management trends that were reported by Cooper et al. in their Fig. 8.21.

Response: Thank you for your suggestions to connect our results with those of Cooper et al. (2022). In response, we added one sentence to the Main text, following:

Lines 197-200: “This result aligns with prior work²⁸, which showed that sequential improvements in hybrid yield under drought were driven by breeding-induced shifts in water-use patterns. Such shifts reduced hybrid sensitivity to drought during the phase of flowering to grain filling.”

Reviewer #1 (Remarks on code availability):

No additional review of code.

Response: Thank you.

-End of Response for Referee 1.

Reviewer #2 (Remarks to the Author):

I praise the authors for considering my suggestions. Incorporating prior literature made this paper complete. Including data from the far west corn belt make this paper much stronger than the prior version. The additional analysis that led to the following conclusions: 1) no changes in VEG but GF, 2) response to VPD underpins DT, 3) there is parallel gain (this was deliberate in AQUAmax breeding objectives), and 4) the percent change between DT and non-DT estimated in this study, are very consistent with what we know about the physiological changes underpinning drought tolerance and provide an independent verification of other studies are cited in the manuscript.

Response: We appreciate the reviewer's insights during the review process, which have improved the quality of the manuscript, and we are grateful for the nice feedback.

-End of Response for Referee 2.

Reviewer #3 (Remarks to the Author):

After going through the entire response and the revised manuscript carefully, I conclude that authors made genuine efforts to address concerns raised by the reviewers. I am satisfied with the explanation in the response and modification in the revised manuscript. This is a very well conceived and completed study, and a significant contribution to the literature!

Response: We sincerely thank the reviewer for the constructive insights during the review process.

A few minor comments:

1. Given the comprehensiveness of this study, it is very helpful to document many detailed analysis and decision steps authors used. Without that, it can even be challenging for authors (maybe except 1st and last author) to repeat all the steps. So, I encourage authors to go through the response document to see whether you can further incorporate any text of reasoning and decision-making text into the supplementary materials. There is also some space (limited in 5,000 words; 3,305) in the main text to accommodate some critical additions. This can be mainly achieved with expanded supplementary texts and supplementary figure captions, if not supplementary methods. This is a generic request to check, not specific parts, so that authors can decide what is best for their paper.

Response: In response to your suggestion, we have added two new Supplementary texts to improve clarity: including (1) “**Text S2. The category of maize hybrids: low-, median-, and high-yielding hybrids**”; and (2) “**Text S4. Procedures to estimate breeding-driven yield gains under varying stress conditions.**”

2. Abstract, Results (L207-227), Figure 5, and related supplementary materials. Please do a targeted reading through to make sure the terms you want to use: old, intermediate, and new; new and new age; and advanced and older.

Response: Thank you for this detailed suggestion. We have standardized the terminology to “old, median, and new” in these subsections.

3. Please think about the title “parallel gains in maize yields and drought tolerance”. Did you want to stress 1) the “parallel” between maize yield and drought tolerance, or 2) “parallel” among different yield groups of hybrids, different environmental conditions, different age groups of hybrids? Good to look at the main messages supported by the main figures and then Abstract to construct a title that is easy to grasp. These 1) and 2) can be somewhat related, but not the same since additional reasoning is needed when you stress in the Abstract, “these gains are accompanied by enhanced drought resistance during the grain filling period”, after “provides evidence of consistent yield increases across diverse environmental conditions.”

Response: Thank you for your insights. Based on your suggestion, we slightly revised title to “Concurrent improvements in maize yield and drought resistance through breeding advances in the U.S. Corn Belt.”

Others:

L631. Modify “The calculated process”.

Response: Yes. We have revised it to “The calculation procedure.”

L638. Rewrite text for panel b.

Response: Thank you. We have rewritten text for panel b, as follows:

Line xx-xx: “**b**, Effect of VPD on yield during the grain filling period (silking to maturity), with sensitivity estimated below and above the VPD threshold as in panel **a**.”

L661. It may be better to omit “(SSP)” here.

Response: Done

-End of Response for Referee 3.